# REGULARIZED INVERSE REINFORCEMENT LEARNING

**Wonseok Jeon**[*,1,2]**, Chen-Yang Su**[1,2]**, Paul Barde**[1,2]**, Thang Doan**[1,2]**,**
**Derek Nowrouzezahrai**[1,2]**, Joelle Pineau**[1,2,3]
[1]Mila, Quebec AI Institute
[2]School of Computer Science, McGill University
[3]Facebook AI Research

## ABSTRACT

Inverse Reinforcement Learning (IRL) aims to facilitate a learner's ability to imitate expert behavior by acquiring reward functions that explain the expert's decisions. *Regularized IRL* applies strongly convex regularizers to the learner's policy in order to avoid the expert's behavior being rationalized by arbitrary constant rewards, also known as degenerate solutions. We propose tractable solutions, and practical methods to obtain them, for regularized IRL. Current methods are restricted to the maximum-entropy IRL framework, limiting them to Shannon-entropy regularizers, as well as proposing solutions that are intractable in practice. We present theoretical backing for our proposed IRL method's applicability to both discrete and continuous controls, empirically validating our performance on a variety of tasks.

## 1 INTRODUCTION

Reinforcement learning (RL) has been successfully applied to many challenging domains including games (Mnih et al., 2015; 2016) and robot control (Schulman et al., 2015; Fujimoto et al., 2018; Haarnoja et al., 2018). Advanced RL methods often employ policy regularization motivated by, e.g., boosting exploration (Haarnoja et al., 2018) or safe policy improvement (Schulman et al., 2015). While Shannon entropy is often used as a policy regularizer (Ziebart et al., 2008), Geist et al. (2019) recently proposed a theoretical foundation of *regularized Markov decision processes* (MDPs)—a framework that uses strongly convex functions as policy regularizers. Here, one crucial advantage is that an optimal policy is shown to *uniquely* exist, whereas multiple optimal policies may exist in the absence of policy regularization.

Meanwhile, since RL requires a given or known reward function (which can often involve non-trivial reward engineering), Inverse Reinforcement Learning (IRL) (Russell, 1998; Ng et al., 2000)—the problem of acquiring a reward function that promotes expert-like behavior—is more generally adopted in practical scenarios like robotic manipulation (Finn et al., 2016b), autonomous driving (Sharifzadeh et al., 2016; Wu et al., 2020) and clinical motion analysis (Li et al., 2018). In these scenarios, defining a reward function beforehand is particularly challenging and IRL is simply more pragmatic. However, complications with IRL in unregularized MDPs relate to the issue of degeneracy, where any constant function can rationalize the expert's behavior (Ng et al., 2000).

Fortunately, Geist et al. (2019) show that IRL in regularized MDPs—*regularized IRL*—does not contain such degenerate solutions due to the uniqueness of the optimal policy for regularized MDPs. Despite this, no tractable solutions of regularized IRL—other than maximum-Shannon-entropy IRL (MaxEntIRL) (Ziebart et al., 2008; Ziebart, 2010; Ho & Ermon, 2016; Finn et al., 2016a; Fu et al., 2018)—have been proposed.

In Geist et al. (2019), solutions for regularized IRL were introduced. However, they are generally intractable since they require a closed-form relation between the policy and optimal value function and the knowledge on model dynamics. Furthermore, practical algorithms for solving regularized IRL problems have not yet been proposed.

We summarize our contributions as follows. Unlike the solutions in Geist et al. (2019), we propose tractable solutions for regularized IRL problems that can be derived from policy regularization and

---

*Correspondence to: Wonseok Jeon `<jeonwons@mila.quebec>`

its gradient in discrete control problems (Section 3.1). We additionally show that our solutions are tractable for Tsallis entropy regularization with multi-variate Gaussian policies in continuous control problems (Section 3.2). We devise Regularized Adversarial Inverse Reinforcement Learning (RAIRL), a practical sample-based method for policy imitation and reward learning in regularized MDPs, which generalizes adversarial IRL (AIRL, Fu et al. (2018)) (Section 4). Finally, we empirically validate our RAIRL method on both discrete and continuous control tasks, evaluating RAIRL via episodic scores and from divergence minimization perspective (Ke et al., 2019; Ghasemipour et al., 2019; Dadashi et al., 2020) (Section 5).

## 2 PRELIMINARIES

**Notation** For finite sets $X$ and $Y$, $Y^X$ is a set of functions from $X$ to $Y$. $\Delta^X$ ($\Delta_Y^X$) is a set of (conditional) probabilities over $X$ (conditioned on $Y$). Especially for the conditional probabilities $p_{X|Y} \in \Delta_Y^X$, we say $p_{X|Y}(\cdot|y) \in \Delta^X$ for $y \in Y$. $\mathbb{R}$ is the set of real numbers. For functions $f_1, f_2 \in \mathbb{R}^X$, we define $\langle f_1, f_2 \rangle_X := \sum_{x \in X} f_1(x) f_2(x)$.

**Regularized Markov Decision Processes and Reinforcement Learning** We consider sequential decision making problems where an agent sequentially chooses its action after observing the state of the environment, and the environment in turn emits a reward with state transition. Such an interaction between the agent and the environment is modeled as an infinite-horizon Markov Decision Process (MDP), $\mathcal{M}^r := \langle \mathcal{S}, \mathcal{A}, P_0, P, r, \gamma \rangle$ and the agent's policy $\pi \in \Delta_{\mathcal{S}}^{\mathcal{A}}$. The terms within the MDP are defined as follows: $\mathcal{S}$ is a finite state space, $\mathcal{A}$ is a finite action space, $P_0 \in \Delta^{\mathcal{S}}$ is an initial state distribution, $P \in \Delta_{\mathcal{S} \times \mathcal{A}}^{\mathcal{S}}$ is a state transition probability, $r \in \mathbb{R}^{\mathcal{S} \times \mathcal{A}}$ is a reward function, and $\gamma \in [0, 1)$ is the discount factor. We also define an MDP without reward as $\mathcal{M}^- := \langle \mathcal{S}, \mathcal{A}, P_0, P, \gamma \rangle$. The normalized state-action visitation distribution, $d_\pi \in \Delta^{\mathcal{S} \times \mathcal{A}}$, associated with $\pi$ is defined as the expected discounted state-action visitation of $\pi$, i.e., $d_\pi(s, a) := (1 - \gamma) \cdot \mathbb{E}_\pi[\sum_{i=0}^\infty \gamma^i \mathbb{I}\{s_i = s, a_i = a\}]$, where the subscript $\pi$ on $\mathbb{E}$ means that a trajectory $(s_0, a_0, s_1, a_1, ...)$ is randomly generated from $\mathcal{M}^-$ and $\pi$, and $\mathbb{I}\{\cdot\}$ is an indicator function. Note that $d_\pi$ satisfies the transposed Bellman recurrence (Boularias & Chaib-Draa, 2010; Zhang et al., 2019):

$$d_\pi(s, a) = (1 - \gamma)P_0(s)\pi(a|s) + \gamma\pi(a|s) \sum_{\bar{s}, \bar{a}} P(s|\bar{s}, \bar{a})d_\pi(s, a).$$

We consider RL in regularized MDPs (Geist et al., 2019), where the policy is optimized with a causal policy regularizer. Mathematically for an MDP $\mathcal{M}^r$ and a strongly convex function $\Omega : \Delta^{\mathcal{A}} \to \mathbb{R}$, the objective in regularized MDPs is to seek $\pi$ that maximizes the expected discounted sum of rewards, or *return* in short, with policy regularizer $\Omega$:

$$\arg\max_{\pi \in \Delta_{\mathcal{S}}^{\mathcal{A}}} J_\Omega(r, \pi) := \mathbb{E}_\pi \left[ \sum_{i=0}^\infty \gamma^i \{r(s_i, a_i) - \Omega(\pi(\cdot|s_i))\} \right] = \frac{1}{1 - \gamma} \mathbb{E}_{(s,a) \sim d_\pi} \left[ r(s, a) - \Omega(\pi(\cdot|s)) \right].$$

$$(1)$$

It turns out that the optimal solution of Eq.(1) is unique (Geist et al., 2019), whereas multiple optimal policies may exist in unregularized MDPs (See Appendix A for a detailed explanation). In later work (Yang et al., 2019), $\Omega(p) = -\lambda\mathbb{E}_{a \sim p}\phi(p(a)), p \in \Delta^{\mathcal{A}}$ was considered for $\lambda > 0$ and $\phi : (0, 1] \to \mathbb{R}$ satisfying some mild conditions. For example, RL with Shannon entropy regularization (Haarnoja et al., 2018) can be recovered by $\phi(x) = -\log x$, while RL with Tsallis entropy regularization (Lee et al., 2020) can be recovered from $\phi(x) = \frac{k}{q-1}(1 - x^{q-1})$ for $k > 0, q > 1$. The optimal policy $\pi^*$ for Eq.(1) with $\Omega$ from Yang et al. (2019) is shown to be

$$\pi^*(a|s) = \max \left\{ g_\phi \left( \frac{\mu^*(s) - Q^*(s, a)}{\lambda} \right), 0 \right\}, \tag{2}$$

$$Q^*(s, a) = r(s, a) + \gamma\mathbb{E}_{s' \sim P(\cdot|s,a)}V^*(s'), V^*(s) = \mu^*(s) - \lambda \sum_{a \in \mathcal{A}} \pi^*(a|s)^2 \phi'(\pi^*(a|s)), \tag{3}$$

where $\phi'(x) = \frac{\partial}{\partial x}\phi(x)$, $g_\phi$ is an inverse function of $f'_\phi$ for $f_\phi(x) := x\phi(x), x \in (0, 1]$, and $\mu^*$ is a normalization term such that $\sum_{a \in \mathcal{A}} \pi^*(a|s) = 1$. Note that we still need to find out $\mu^*$ to acquire a closed-form relation between optimal policy $\pi^*$ and value function $Q^*$. However, such relations

have not been discovered except for Shannon-entropy regularization (Haarnoja et al., 2018) and specific instances ($q = 1, 2, \infty$) of Tsallis-entropy regularization (Lee et al., 2019) to the best of our knowledge.

**Inverse Reinforcement Learning**   Given a set of demonstrations from an expert policy $\pi_E$, IRL (Russell, 1998; Ng et al., 2000) is the problem of seeking a reward function from which we can recover $\pi_E$ through RL. However, IRL in unregularized MDPs has been shown to be an ill-defined problem since *(1)* any constant reward function can rationalize every expert and *(2)* multiple rewards meet the criteria of being a solution (Ng et al., 2000). Maximum entropy IRL (MaxEntIRL) (Ziebart et al., 2008; Ziebart, 2010) is capable of solving the first issue by seeking a reward function that maximizes the expert's return along with Shannon entropy of expert policy. Mathematically, for the RL objective $J_\Omega$ in Eq.(1) and $\Omega = -\mathcal{H}$ for negative Shannon entropy $\mathcal{H}(p) = \mathbb{E}_{a \sim p}[-\log p(a)]$ (Ho & Ermon, 2016), the objective of MaxEntIRL is

$$\text{MaxEntIRL}(\pi_E) := \underset{r \in \mathbb{R}^{S \times A}}{\arg\max} \left\{ J_{-\mathcal{H}}(r, \pi_E) - \max_{\pi \in \Delta_S^A} J_{-\mathcal{H}}(r, \pi) \right\}. \tag{4}$$

Another commonly used IRL method is Adversarial Inverse Reinforcement Learning (AIRL) (Fu et al., 2018) which involves generative adversarial training (Goodfellow et al., 2014; Ho & Ermon, 2016) to acquire a solution of MaxEntIRL. AIRL considers the structured discriminator (Finn et al., 2016a) $D(s, a) = \sigma(r(s, a) - \log \pi(a|s)) = \frac{e^{r(s,a)}}{e^{r(s,a)} + \pi(a|s)}$ for $\sigma(x) := 1/(1 + e^{-x})$ and iteratively optimizes the following objective:

$$\max_{r \in \mathbb{R}^{S \times A}} \mathbb{E}_{(s,a) \sim d_{\pi_E}} \left[ \log D_{r,\pi}(s, a) \right] + \mathbb{E}_{(s,a) \sim d_\pi} \left[ \log(1 - D_{r,\pi}(s, a)) \right],$$

$$\max_{\pi \in \Delta_S^A} \mathbb{E}_{(s,a) \sim d_\pi} \left[ \log D_{r,\pi}(s, a) - \log(1 - D_{r,\pi}(s, a)) \right] = \max_{\pi \in \Delta_S^A} \mathbb{E}_{(s,a) \sim d_\pi} \left[ r(s, a) - \log \pi(a|s) \right]. \tag{5}$$

It turns out that AIRL minimizes the divergence between visitation distributions $d_\pi$ and $d_{\pi_E}$ by solving $\min_{\pi \in \Delta_S^A} \text{KL}(d_\pi || d_{\pi_E})$ for Kullback-Leibler (KL) divergence $\text{KL}(\cdot || \cdot)$ (Ghasemipour et al., 2019) .

## 3   INVERSE REINFORCEMENT LEARNING IN REGULARIZED MDPS

In this section, we propose the solution of IRL in regularized MDPs—*regularized IRL*—and relevant properties in Section 3.1. We then discuss a specific instance of our proposed solution in Section 3.2 where Tsallis entropy regularizers and multi-variate Gaussian policies are used in continuous action spaces.

### 3.1   SOLUTIONS OF REGULARIZED IRL

We consider regularized IRL that generalizes MaxEntIRL in Eq.(4) to IRL with a class of strongly convex policy regularizers:

$$\text{IRL}_\Omega(\pi_E) := \underset{r \in \mathbb{R}^{S \times A}}{\arg\max} \left\{ J_\Omega(r, \pi_E) - \max_{\pi \in \Delta_S^A} J_\Omega(r, \pi) \right\}. \tag{6}$$

For any strongly convex policy regularizer $\Omega$, regularized IRL does not suffer from degenerate solutions since there is a unique optimal policy in any regularized MDP (Geist et al., 2019). While Geist et al. (2019) proposed solutions of regularized IRL, those are intractable solutions (See Appendix F.1 for a detailed explanation). In the following lemma, we propose tractable solutions that only requires the evaluation of the policy regularizer ($\Omega$) and its gradient ($\nabla \Omega$) which are more manageable in practice. Our solution is motivated from figuring out a reward function that is capable of converting regularized RL into an equivalent divergence minimization problem associated with $\pi$ and $\pi_E$:

**Lemma 1.** *For a policy regularizer $\Omega : \Delta^A \to \mathbb{R}$, let us define*

$$t(s, a; \pi) := \Omega'(s, a; \pi) - \mathbb{E}_{a' \sim \pi(\cdot|s)}[\Omega'(s, a'; \pi)] + \Omega(\pi(\cdot|s)) \tag{7}$$

*for $\Omega'(s, \cdot; \pi) := \nabla \Omega(\pi(\cdot|s)) := [\nabla_p \Omega(p)]_{p=\pi(\cdot|s)} \in \mathbb{R}^A, s \in \mathcal{S}$. Then, $t(s, a; \pi_E)$ for expert policy $\pi_E$ is a solution of regularized IRL with $\Omega$.*

*Proof.* (Abbreviated. See Appendix B for the full version.) With $r(s,a) = t(s,a;\pi_E)$, the RL objective in Eq.(1) becomes equivalent to *the problem of minimizing the discounted sum of Bregman divergences between $\pi$ and $\pi_E$*

$$\arg\min_{\pi\in\Delta_S^A} \mathbb{E}_\pi\left[\sum_{i=0}^\infty \gamma^i D_\Omega^A(\pi(\cdot|s_i)||\pi_E(\cdot|s_i))\right], \tag{8}$$

where $D_\Omega^A$ is the Bregman divergence (Bregman, 1967) defined by $D_\Omega^A(p_1||p_2) = \Omega(p_1) - \Omega(p_2) - \langle\nabla\Omega(p_2), p_1 - p_2\rangle_A$ for $p_1, p_2 \in \Delta^A$. Due to the non-negativity of the Bregman divergence, $\pi = \pi_E$ is a solution of Eq.(8) and is unique since Eq.(1) has the unique solution for arbitrary reward functions (Geist et al., 2019). $\square$

In particular, for any policy regularizer $\Omega$ represented by an expectation over the policy (Yang et al., 2019), **Lemma 1** can be reduced to the following solution in **Corollary 1**:

**Corollary 1.** *For $\Omega(p) = -\lambda\mathbb{E}_{a\sim p}\phi(p(a))$ with $p \in \Delta^A$ (Yang et al., 2019), Eq.(7) becomes*

$$t(s,a;\pi) = -\lambda\cdot\left\{f_\phi'(\pi(a|s)) - \mathbb{E}_{a'\sim\pi(\cdot|s)}[f_\phi'(\pi(a'|s)) - \phi(\pi(a'|s))]\right\} \tag{9}$$

*for $f_\phi'(x) = \frac{\partial}{\partial x}(x\phi(x))$. The proof is in Appendix B.*

Throughout the paper, we denote **reward baseline** by the expectation $\mathbb{E}_{a\sim\pi(\cdot|s)}[f_\phi'(\pi(a|s)) - \phi(\pi(a|s))]$. Note that for *continuous control tasks* with $\Omega(p) = -\lambda\mathbb{E}_{a\sim p}\phi(p(a))$, we can obtain the same form of the reward in Eq.(9) (The proof is in Appendix D). Although the reward baseline is generally not intractable in continuous control tasks, we derive a tractable reward baseline for a special case (See Section 3.2). Additionally, when $\lambda = 1$ and $\phi(x) = -\log x$, it can be shown that $t(s,a;\pi) = \log\pi(a|s)$—for both discrete and continuous control problems—which was used as a reward objective in previous work (Fu et al., 2018), and that the Bregman divergence in Eq.(8) becomes the KL divergence $\mathrm{KL}(\pi(\cdot|s)||\pi_E(\cdot|s))$.

Additional solutions of regularized IRL can be found by shaping $t(s,a;\pi_E)$ as stated in the following lemma:

**Lemma 2** (Potential-based reward shaping). *Let $\pi^*$ be the solution of Eq.(1) in a regularized MDP $\mathcal{M}^r$ with a regularizer $\Omega : \Delta^A \to \mathbb{R}$ and a reward function $r \in \mathbb{R}^{S\times A}$. Then for $\Phi \in \mathbb{R}^S$, using either $r(s,a) + \gamma\Phi(s') - \Phi(s)$ or $r(s,a) + \mathbb{E}_{s'\sim P(\cdot|s,a)}\Phi(s') - \Phi(s)$ as a reward does not change the solution of Eq.(1). The proof is in Appendix E.*

From **Lemma 1** and **Lemma 2**, we prove the sufficient condition of rewards being solutions of the IRL problem. However, the necessary condition—a set of those solutions are the only possible solutions for an arbitrary MDP—is not proved (Ng et al., 1999).

In the following lemma, we check how the proposed solution is related to the normalized state-visitation distribution which can be discussed in the line of distribution matching perspective on imitation learning problems (Ho & Ermon, 2016; Fu et al., 2018; Ke et al., 2019; Ghasemipour et al., 2019):

**Lemma 3.** *Given the policy regularizer $\Omega$, let us define $\bar{\Omega}(d) := \mathbb{E}_{(s,a)\sim d}[\Omega(\bar{\pi}_d(\cdot|s))]$ for an arbitrary normalized state-action visitation distribution $d \in \Delta^{S\times A}$ and the policy $\bar{\pi}_d(a|s) := \frac{d(s,a)}{\sum_{a'\in A}d(s,a')}$ induced by $d$. Then, Eq.(7) is equal to*

$$t(s,a;\bar{\pi}_d) = [\nabla\bar{\Omega}(d)](s,a). \tag{10}$$

*When $\bar{\Omega}(d)$ is strictly convex and a solution $t(s,a;\pi_E) = \nabla\bar{\Omega}(d_{\pi_E})$ of IRL in Eq.(10) is used, the RL objective in Eq.(1) is equal to*

$$\arg\min_{\pi\in\Delta_S^A} D_{\bar{\Omega}}^{S\times A}(d_\pi||d_{\pi_E}),$$

*where $D_{\bar{\Omega}}^{S\times A}$ is the Bregman divergence among visitation distributions defined by $D_{\bar{\Omega}}^{S\times A}(d_1||d_2) = \bar{\Omega}(d_1) - \bar{\Omega}(d_2) - \langle\nabla\bar{\Omega}(d_2), d_1 - d_2\rangle$ for visitation distributions $d_1$ and $d_2$. The proof is in Appendix G.*

Note that the strict convexity of $\bar{\Omega}$ is required for $D_{\bar{\Omega}}^{\mathcal{S} \times \mathcal{A}}$ to become a valid Bregman divergence. Although the strict convexity of a policy regularizer $\bar{\Omega}$ does not guarantee the assumption on the strict convexity of $\bar{\Omega}$, it has been shown to be true for Shannon entropy regularizer ($\bar{\Omega} = -\bar{H}$ of **Lemma 3.1** in Ho & Ermon (2016)) and Tsallis entropy regularizer with its constants $k = \frac{1}{2}, q = 2$ ($\bar{\Omega} = -\bar{W}$ of **Theorem 3** in Lee et al. (2018)).

## 3.2 IRL with Tsallis entropy regularization and Gaussian policies

For continuous controls, multi-variate Gaussian policies are often used in practice (Schulman et al., 2015; 2017) and we consider IRL problems with those policies in this subsection. In particular, we consider IRL with Tsallis entropy regularizer $\Omega(p) = -\mathcal{T}_q^k(p) = -\mathbb{E}_{a \sim p}[\frac{k}{q-1}(1 - p(a)^{q-1})]$ (Lee et al., 2018; Yang et al., 2019; Lee et al., 2020) for a multi-variate Gaussian policy $\pi(\cdot|s) = \mathcal{N}(\boldsymbol{\mu}(s), \boldsymbol{\Sigma}(s))$ with $\boldsymbol{\mu}(s) = [\mu_1(s), ..., \mu_d(s)]^T$, $\boldsymbol{\Sigma}(s) = \text{diag}\{(\sigma_1(s))^2, ..., (\sigma_d(s))^2\}$. In such a case, we can obtain tractable forms of the following quantities:

**Tsallis entropy.** The tractable form of Tsallis entropy for a multi-variate Gaussian policy is

$$\mathcal{T}_q^k(\pi(\cdot|s)) = \frac{k(1 - e^{(1-q)\mathcal{R}_q(\pi(\cdot|s))})}{q-1}, \mathcal{R}_q(\pi(\cdot|s)) = \sum_{i=1}^{d}\left\{\log(\sqrt{2\pi}\sigma_i(s)) - \frac{\log q}{2(1-q)}\right\}$$

for Renyi entropy $\mathcal{R}_q$. Its derivation is given in Appendix I.

**Reward baseline.** The reward baseline term $\mathbb{E}_{a \sim \pi(\cdot|s)}[f'_\phi(\pi(a|s)) - \phi(\pi(a|s))]$ in **Corollary 1** is generally intractable except for either discrete control problems or Shannon entropy regularization where the reward baseline is equal to $-1$. Interestingly, as long as the tractable form of Tsallis entropy can be derived, that of the corresponding reward baseline can also be derived since the reward baseline satisfies

$$\mathbb{E}_{a \sim \pi(\cdot|s)}[f'_\phi(\pi(a|s)) - \phi(\pi(a|s))] = (q-1)\mathbb{E}_{a \sim \pi(\cdot|s)}[\phi(\pi(a|s)] - k = (q-1)\mathcal{T}_q^k(\pi(\cdot|s)) - k.$$

Here, the first equality holds with $f'_\phi(x) = \frac{k}{q-1}(1 - qx^{q-1}) = q\phi(x) - k$ for Tsallis entropy regularization.

**Bregman divergence associated with Tsallis entropy.** For two different multivariate Gaussian policies, we derive the tractable form of the Bregman divergence (associated with Tsallis entropy) between two policies. The resultant divergence has a complicated form, so we leave its derivation in Appendix I.3.

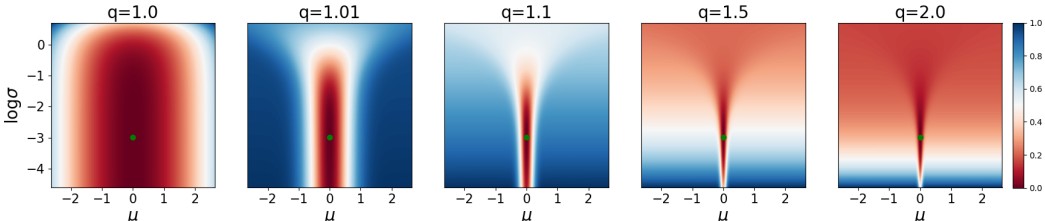

Figure 1: Bregman divergence $D_\Omega(\pi||\pi_E)$ associated with Tsallis entropy ($\Omega(p) = -\mathcal{T}_q^1(p)$) between two uni-variate Gaussian distributions $\pi = \mathcal{N}(\mu, \sigma^2)$ and $\pi_E = \mathcal{N}(0, (e^{-3})^2)$ (green point in each subplot). In each subplot, we normalized the Bregman divergence so that the maximum value becomes 1. Note that for $q = 1$, $D_\Omega(\pi||\pi_E)$ becomes the KL divergence $\text{KL}(\pi||\pi_E)$.

For deeper understanding of Tsallis entropy and its associated Bregman divergence in regularized MDPs, we consider an example in Figure 1. We first assume that both learning agents' and experts' policies follow uni-variate Gaussian distributions $\pi = \mathcal{N}(\mu, \sigma^2)$ and $\pi_E = \mathcal{N}(0, (e^{-3})^2)$, respectively. We then evaluate the Bregman divergence in Figure 1 by using its tractable form and varying $q$ from 1.0—which corresponds to the KL divergence—to 2.0. We observe that the constant $q$ from the Tsallis entropy affects the sensitivity of the associated Bregman divergence w.r.t. the mean and standard deviation of the learning agent's policy $\pi$. Specifically, as $q$ increases, the size of the valley—relatively red region in Figure 1—across the $\mu$-axis and $\log \sigma$-axis decreases. This suggests that for larger $q$, minimizing the Bregman divergence requires more tightly matching means and variances of $\pi$ and $\pi_E$.

---

**Algorithm 1:** Regularized Adversarial Inverse Reinforcement Learning (RAIRL)

---

1: **Input:** A set $\mathcal{D}_E$ of expert demonstration generated by expert policy $\pi_E$, a reward approximator $r_\theta$, a policy $\pi_\psi$ for neural network parameters $\theta$ and $\psi$

2: **for** each iteration **do**

3:     Sample rollout trajectories by using the learners policy $\pi_\psi$.

4:     Optimize $\theta$ with the discriminator $D_{r_\theta, \pi_\psi}$ and the learning objective in Eq.(11).

5:     Optimize $\psi$ with Regularized Actor Critic by using $r_\theta$ as a reward function.

6: **end for**

7: **Output:** $\pi_\psi \approx \pi_E$, $r_\theta(s,a) \approx t(s,a;\pi_E)$ (a solution of the IRL problem in **Lemma 1**).

---

## 4 ALGORITHMIC CONSIDERATION

Based on a solution for regularized IRL in the previous section, we focus on developing an IRL algorithm in this section. Particularly to recover the reward function $t(s,a;\pi_E)$ in **Lemma 1**, we design an adversarial training objective as follows. Motivated by AIRL (Fu et al., 2018), we consider the following structured discriminator associated with $\pi, r$ and $t$ in **Lemma 1**:

$$D_{r,\pi}(s,a) = \sigma(r(s,a) - t(s,a;\pi)), \sigma(z) = \frac{1}{1+e^{-z}}, z \in \mathbb{R}.$$

Note that we can recover the discriminator of AIRL in Eq.(5) when $t(s,a) = \log \pi(a|s)$ ($\phi(x) = \log x$ and $\lambda = 1$). Then, we consider the following optimization objective of the discriminator which is the same as that of AIRL:

$$\hat{t}(s,a;\pi) := \underset{r \in \mathbb{R}^{\mathcal{S} \times \mathcal{A}}}{\arg\max} \; \mathbb{E}_{(s,a) \sim d_{\pi_E}} \left[ \log D_{r,\pi}(s,a) \right] + \mathbb{E}_{(s,a) \sim d_\pi} \left[ \log(1 - D_{r,\pi}(s,a)) \right]. \quad (11)$$

Since the function $x \mapsto a \log \sigma(x) + b \log(1 - \sigma(x))$ attains its maximum at $\sigma(x) = \frac{a}{a+b}$, or equivalently at $x = \log \frac{a}{b}$ (Goodfellow et al., 2014; Mescheder et al., 2017), it can be shown that

$$\hat{t}(s,a;\pi) = t(s,a;\pi) + \log \frac{d_{\pi_E}(s,a)}{d_\pi(s,a)}. \quad (12)$$

When $\pi = \pi_E$ in Eq.(12), we have $\hat{t}(s,a;\pi_E) = t(s,a;\pi_E)$ since $d_\pi = d_{\pi_E}$, which means the maximizer $\hat{t}$ becomes the solution of IRL *after* the agent successfully imitates the expert policy $\pi_E$. To do so, we consider the following iterative algorithm. Assuming that we find out the optimal reward approximator $\hat{t}(s,a;\pi^{(i)})$ in Eq.(12) for the policy $\pi^{(i)}$ of the $i$-th iteration, we get the policy $\pi^{(i+1)}$ by optimizing the following objective with gradient ascent:

$$\underset{\pi \in \Delta_{\mathcal{S}}^{\mathcal{A}}}{\text{maximize}} \; \mathbb{E}_{(s,a) \sim d_\pi} \left[ \hat{t}(s,a;\pi^{(i)}) - \Omega(\pi(\cdot|s)) \right]. \quad (13)$$

The above expectation in Eq.(13) can be decomposed into the following two terms

$$\mathbb{E}_{(s,a) \sim d_\pi} \left[ \hat{t}(s,a;\pi^{(i)}) - \Omega(\pi(\cdot|s)) \right] = \mathbb{E}_{(s,a) \sim d_\pi} \left[ t(s,a;\pi^{(i)}) - \Omega(\pi(\cdot|s)) \right] - \mathrm{KL}(d_\pi || d_{\pi_E})$$

$$= -\underbrace{\mathbb{E}_{(s,a) \sim d_\pi} \left[ D_\Omega^{\mathcal{A}}(\pi(\cdot|s) || \pi^{(i)}(\cdot|s)) \right]}_{(\mathrm{I})} - \underbrace{\mathrm{KL}(d_\pi || d_{\pi_E})}_{(\mathrm{II})}, \quad (14)$$

where the second equality follows since **Lemma 1** tells us that $t(s,a;\pi^{(i)})$ is a reward function that makes $\pi^{(i)}$ an optimal policy in the $\Omega$-regularized MDP. Minimizing term (II) in Eq.(14) makes $\pi^{(i+1)}$ close to $\pi_E$ while minimizing term (I) can be regarded as a conservative policy optimization around the policy $\pi^{(i)}$ (Schulman et al., 2015).

In practice, we parameterize our reward and policy approximations with neural networks and train them using an off-policy Regularized Actor-Critic (RAC) (Yang et al., 2019) as described in **Algorithm 1**. Below, we evaluate our Regularized Adversarial Inverse Reinforcement Learning (RAIRL) approach across various scenarios.

## 5 EXPERIMENTS

We summarize the experimental setup as follows. In our experiments, we consider $\Omega(p) = -\lambda\mathbb{E}_{a\sim p}[\phi(p(a)]$ with the following regularizers from Yang et al. (2019): *(1) Shannon entropy* ($\phi(x) = -\log x$), *(2) Tsallis entropy regularizer* ($\phi(x) = \frac{k}{q-1}(1 - x^{q-1})$), *(3)* exp *regularizer* ($\phi(x) = e - e^x$), *(4)* cos *regularizer* ($\phi(x) = \cos(\frac{\pi}{2}x)$), *(5)* sin *regularizer* ($\phi(x) = 1 - \sin\frac{\pi}{2}x$). The above regularizers were chosen since other regularizers have not been empirically validated to the best of our knowledge. We chose these regularizers to make our empirical analysis more tractable. In addition, we model the reward approximator of RAIRL as a neural network with either one of the following models: *(1) Non-structured model (NSM)*—a simple feed-forward neural network that outputs real values used in AIRL (Fu et al., 2018)—and *(2) Density-based model (DBM)*—a model using a neural network for $\pi$ (softmax for discrete controls and multi-variate Gaussian model for continuous controls) of the solution in Eq.(1) (See Appendix J.2 for a detailed explanation). For the RL algorithm of RAIRL, we implement Regularized Actor Critic (RAC) (Yang et al., 2019) on top of the SAC implementation from Rlpyt (Stooke & Abbeel, 2019). Other settings are summarized in Appendix J. For all experiments, we use 5 runs and report 95% confidence intervals.

### 5.1 EXPERIMENT 1: MULTI-ARMED BANDIT (DISCRETE ACTION)

We consider a 4-armed bandit environment as shown in Figure 2 (left). An expert policy $\pi_E$ is assumed to be either *dense* (with probability 0.1, 0.2, 0.3, 0.4 for $a = 0, 1, 2, 3$) or *sparse* (with probability 0, 0, 1/3, 2/3 for $a = 0, 1, 2, 3$). For those experts, we use RAIRL with actions sampled from $\pi_E$ and compare learned rewards with the ground truth reward $t(s, a; \pi_E)$ in **Lemma 1**. When $\pi_E$ is dense, RAIRL successfully acquires the ground truth rewards irrespective of the reward model choices. When sparse $\pi_E$ is used, however, RAIRL with a non-structured model (RAIRL-NSM) fails to recover the rewards for $a = 0, 1$—where $\pi_E(a) = 0$—due to the lack of samples at the end of the imitation. On the other hand, RAIRL with a density-based model (RAIRL-DBM) can recover the correct rewards due to the softmax layer which maintains the sum over the outputs equal to 1. Therefore, we argue that using DBM is necessary for correct reward acquisition since a set of demonstrations is generally sparse. In the following experiment, we show that the choice of reward models indeed affects the performance of rewards.

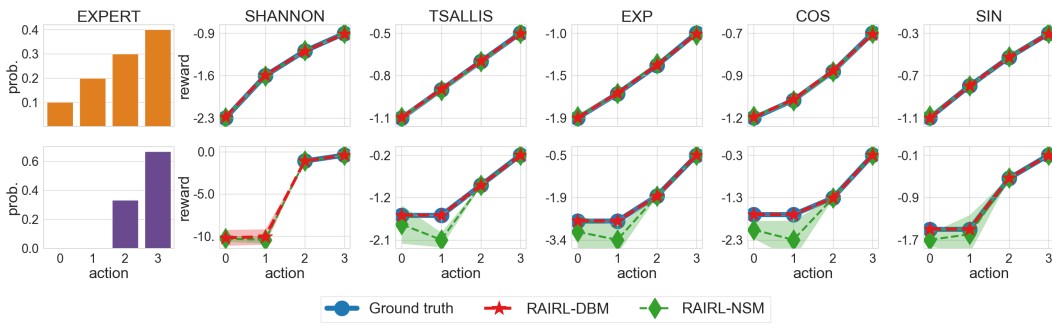

Figure 2: Expert policy (*Left*) and reward learned by RAIRL with different types of policy regularizers (*Right*) in Multi-armed Bandit. Either one of dense (*Top row*) or sparse (*Bottom row*) expert policies $\pi_E$ is considered.

### 5.2 EXPERIMENT 2: BERMUDA WORLD (CONTINUOUS OBSERVATION, DISCRETE ACTION)

We consider an environment with a 2-dimensional continuous state space as described in Figure 3. At each episode, the learning agent is initialized uniformly on the $x$-axis between $-5$ and $5$, and there are 8 possible actions—an angle in $\{-\pi, -\frac{3}{4}\pi, ..., \frac{1}{2}\pi, \frac{3}{4}\pi\}$ that determines the direction of movement. An expert in Bermuda World considers 3 target positions $(-5, 10), (0, 10), (5, 10)$ and behaves stochastically. We state how we mathematically define the expert policy $\pi_E$ in Appendix J.3. During RAIRL's training (Figure 3, *Top row*), we use 1000 demonstrations sampled from the expert and periodically measure mean Bregman divergence, i.e., for $D_\Omega^\mathcal{A}(p_1\|p_2) = \mathbb{E}_{a\sim p_1}[f'_\phi(p_2(a)) -$

$$\phi(p_1(a))] - \mathbb{E}_{a \sim p_2}[f'_\phi(p_2(a)) - \phi(p_2(a))],$$

$$\frac{1}{N}\sum_{i=1}^{N} D^{\mathcal{A}}_{\Omega}(\pi(\cdot|s_i)||\pi_E(\cdot|s_i)).$$

Here, the states $s_1, ..., s_N$ come from 30 evaluation trajectories that are stochastically sampled from the agent's policy $\pi$—which is fixed during evaluation—in a separate evaluation environment. During the evaluation of learned reward (Figure 3, *Bottom row*), we train randomly initialized agents with RAC and rewards acquired from RAIRL's training and check if the mean Bregman divergence is properly minimized. We measure the *mean Bregman divergence* as was done in RAIRL's training.

RAIRL-DBM is shown to minimize the target divergence more effectively compared to RAIRL-NSM during reward evaluation, although both achieve comparable performances during RAIRL's training. Moreover, we substitute $\lambda$ with $1, 5, 10$ and observe that learning with $\lambda$ larger than 1 returns better rewards—only $\lambda = 1$ was considered in AIRL (Fu et al., 2018). Note that in all cases, the minimum divergence achieved by RAIRL is comparable with that of behavioral cloning (BC). This is because BC performs sufficiently well when many demonstrations are given. We think the divergence of BC may be the near-optimal divergence that can be achieved with our policy neural network model.

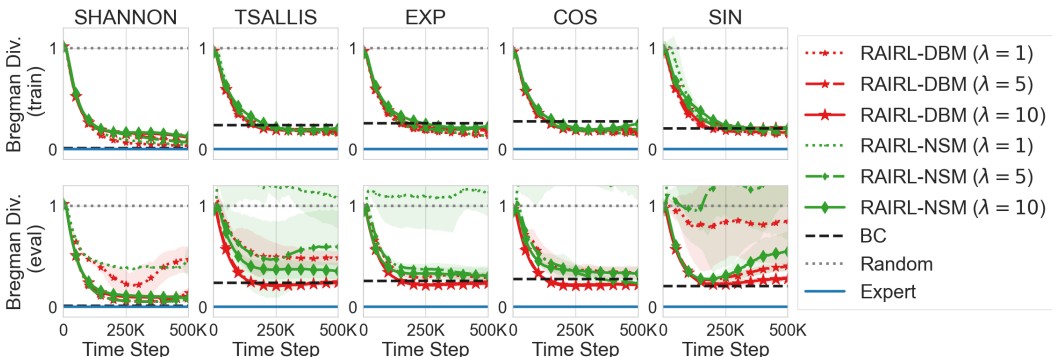

Figure 3: Mean Bregman divergence during training (*Top row*) and the divergence during reward evaluation (*Bottom row*) in Bermuda World. In each column, different policy regularizers and their respective target divergences are considered. The results are reported after normalization with the divergence of uniform random policy, and that of behavioral cloning (BC) is reported for comparison.

### 5.3 EXPERIMENT 3: MUJOCO (CONTINUOUS OBSERVATION AND ACTION)

We validate RAIRL on MuJoCo continuous control tasks (*Hopper-v2*, *Walker-v2*, *HalfCheetah-v2*, *Ant-v2*) as follows. We assume multivariate-Gaussian policies (with diagonal covariance matrices) for both learner's policy $\pi$ and expert policy $\pi_E$. Instead of $\tanh$-squashed policy in Soft-Actor Critic (Haarnoja et al., 2018), we use hyperbolized environments—where $\tanh$ is regarded as a part of the environment—with additional engineering on the policy networks (See Appendix J.4 for details). We use 100 demonstrations stochastically sampled from $\pi_E$ to validate RAIRL. In MuJoCo experiments, we focus on a set of Tsallis entropy regularizer ($\Omega = -\mathcal{T}^1_q$) with $q = 1, 1.5, 2$—where Tsallis entropy becomes Shannon entropy for $q = 1$. We then exploit the tractable quantities for multi-variate Gaussian distributions in Section 3.2 to stabilize RAIRL and check its performance in terms of its mean Bregman divergence similar to the previous experiment. Note that since both $\pi$ and $\pi_E$ are multi-variate Gaussian and can be evaluated, we can evaluate the individual Bregman divergence $D^{\mathcal{A}}_{\Omega}(\pi(\cdot|s)||\pi_E(\cdot|s))$ on $s$ by using the derivation in Appendix I.3.

The performances during RAIRL's training are described as follows. We report $\pi$ with both an episodic score (Figure 4) and mean Bregman divergences with respect to three types of Tsallis entropies (Figure 5) $\Omega = -\mathcal{T}^1_{q'}$ with $q' = 1, 1.5, 2$. Note that the objective of RAIRL with $\Omega = -\mathcal{T}^1_q$ is to minimize the corresponding mean Bregman divergence with $q' = q$. In Figure 4, both RAIRL-DBM and RAIRL-NSM are shown to achieve the expert performance, irrespective of $q$, in *Hopper-v2*, *Walker-v2*, and *HalfCheetah-v2*. In contrast, RAIRL in *Ant-v2* fails to achieve the expert's performance within 2,000,000 steps and RAIRL-NSM highly outperforms RAIRL-DBM in our

setting. Although the episodic scores are comparable for all methods in *Hopper-v2*, *Walker-v2*, and *HalfCheetah-v2*, respective divergences are shown to be highly different from one another as shown in Figure 5. RAIRL with $q = 2$ in most cases achieves the minimum mean Bregman divergence (for all three divergences with $q' = 1, 1.5, 2$), whereas RAIRL with $q = 1$—which corresponds to AIRL (Fu et al., 2018)—achieves the maximum divergence in most cases. This result is in alignment with our intuition from Section 3.2; as $q$ increases, minimizing the Bregman divergence requires much tighter matching between $\pi$ and $\pi_E$. Unfortunately, while evaluating the acquired reward—RAC with a randomly initialized agent and acquired reward—the target divergence is not properly decreased in continuous controls. We believe this is because $\pi$ is a probability density function in continuous controls and causes large variance during training, while $\pi$ is a mass function and is well-bounded in discrete control problems.

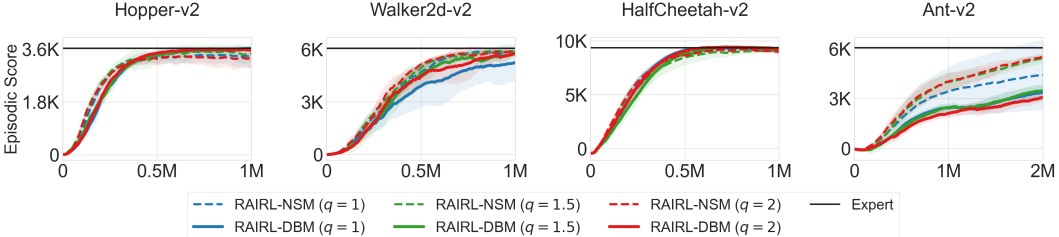

Figure 4: Averaged episodic score of RAIRL's training in MuJoCo environments. RAIRL with $\mathcal{T}_q^1$ regularizer with $q = 1, 1.5, 2$ is considered.

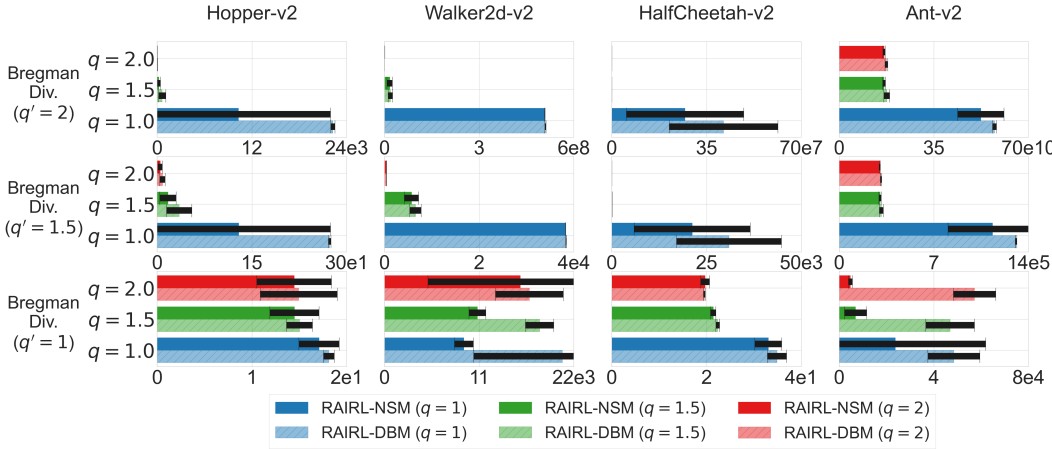

Figure 5: Bregman divergences with Tsallis entropy $\mathcal{T}_{q'}^1$ with $q' = 1, 1.5, 2$ during RAIRL's training in MuJoCo environments. We consider RAIRL with Tsallis entropy regularizer $\mathcal{T}_q^1$ with $q = 1, 1.5, 2$.

# 6 DISCUSSION AND FUTURE WORKS

We consider the problem of IRL in regularized MDPs (Geist et al., 2019), assuming a class of strongly convex policy regularizers. We theoretically derive its solution (a set of reward functions) and show that learning with these rewards is equivalent to a specific instance of imitation learning—i.e., one that minimizes the Bregman divergence associated with policy regularizers. We propose RAIRL—a practical sampled-based IRL algorithm in regularized MDPs—and evaluate its applicability on policy imitation (for discrete and continuous controls) and reward acquisition (for discrete control).

Finally, recent advances in imitation learning and IRL are built from the perspective of regarding imitation learning as statistical divergence minimization problems (Ke et al., 2019; Ghasemipour et al., 2019). Although Bregman divergence is known to cover various divergences, it does not include some divergence families such as $f$-divergence (Csiszár, 1963; Amari, 2009). Therefore, we believe that considering RL with policy regularization different from Geist et al. (2019) and its inverse problem is a possible way of finding the links between imitation learning and various statistical distances.

ACKNOWLEDGMENTS

We would like to thank Daewoo Kim at Waymo for his valuable comments on this work.

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

## A  BELLMAN OPERATORS, VALUE FUNCTIONS IN REGULARIZED MDPs

Let a policy regularizer $\Omega : \Delta^{\mathcal{A}} \to \mathbb{R}$ be strongly convex, and define the convex conjugate of $\Omega$ is $\Omega^* : \mathbb{R}^{\mathcal{A}} \to \mathbb{R}$ as

$$\Omega^*(Q(s, \cdot)) = \max_{\pi(\cdot|s) \in \Delta^{\mathcal{A}}} \langle \pi(\cdot|s), Q(s, \cdot) \rangle_{\mathcal{A}} - \Omega(\pi(\cdot|s)), Q \in \mathbb{R}^{\mathcal{S} \times \mathcal{A}}, s \in \mathcal{S}. \tag{15}$$

Then, Bellman operators, equations and value functions in regularied MDPs are defined as follows.

**Definition 1** (Regularized Bellman Operators). *For $V \in \mathbb{R}^{\mathcal{S}}$, let us define $Q(s, a) = r(s, a) + \gamma \mathbb{E}_{s' \sim P(\cdot|s,a)}[V(s')]$. The regularized Bellman evaluation operator is defined as*

$$[T^\pi V](s) := \langle \pi(\cdot|s), Q(s, \cdot) \rangle_{\mathcal{A}} - \Omega(\pi(\cdot|s)), s \in \mathcal{S},$$

*and $T^\pi V = V$ is called the regularized Bellman equation. Also, the regularized Bellman optimality operator is defined as*

$$[T^* V](s) := \max_{\pi(\cdot|s) \in \Delta^{\mathcal{A}}} [T^\pi V](s) = \Omega^*(Q(s, \cdot)), s \in \mathcal{S},$$

*and $T^* V = V$ is called the regularized Bellman optimality equation.*

**Definition 2** (Regularized value functions). *The unique fixed point $V^\pi$ of the operator $T^\pi$ is called the state value function, and $Q^\pi(s, a) = r(s, a) + \gamma \mathbb{E}_{s' \sim P(\cdot|s,a)}[V^\pi(s')]$ is called the state-action value function. Also, the unique fixed point $V^*$ of the operator $T^*$ is called the optimal state value function, and $Q^*(s, a) = r(s, a) + \gamma \mathbb{E}_{s' \sim P(\cdot|s,a)}[V^*(s')]$ is called the optimal state-action value function.*

It should be noted that **Proposition 1** in Geist et al. (2019) tells us $\nabla \Omega^*(Q(s, \cdot))$ is a policy that *uniquely* maximizes Eq.(15). For example, when $\Omega(\pi(\cdot|s)) = \sum_{a \sim \pi(\cdot|s)} \log \pi(a|s)$ (negative Shannon entropy), $\nabla \Omega^*(Q(s, \cdot))$ is a softmax policy, i.e., $\nabla \Omega^*(Q(s, \cdot)) = \frac{\exp(Q(s, \cdot))}{\sum_{a' \in \mathcal{A}} \exp(Q(s, a'))}$. Due to this property, the optimal policy $\pi^*$ of a regularized MDP is uniquely found for the optimal state-action value function $Q^*$ which is also uniquely defined as the fixed point of $T^*$.

## B  PROOF OF **LEMMA 1**

Let us define $\pi^s = \pi(\cdot|s)$. For $r(s, a) = t(s, a; \pi_E)$, the RL objective Eq.(1) satisfies

$$\mathbb{E}_\pi \left[ \sum_{i=0}^\infty \gamma^i \{ r(s_i, a_i) - \Omega(\pi^{s_i}) \} \right] \stackrel{(i)}{=} \mathbb{E}_\pi \left[ \sum_{i=0}^\infty \gamma^i \{ t(s, a; \pi_E) - \Omega(\pi^{s_i}) \} \right]$$

$$\stackrel{(ii)}{=} \mathbb{E}_\pi \left[ \sum_{i=0}^\infty \gamma^i \left\{ \Omega'(s_i, a_i; \pi_E) - \mathbb{E}_{a \sim \pi_E^{s_i}} \Omega'(s_i, a; \pi_E) + \Omega(\pi_E^{s_i}) - \Omega(\pi^{s_i}) \right\} \right]$$

$$\stackrel{(iii)}{=} \mathbb{E}_\pi \left[ \sum_{i=0}^\infty \gamma^i \left\{ \mathbb{E}_{a \sim \pi^{s_i}} \Omega'(s_i, a; \pi_E) - \mathbb{E}_{a \sim \pi_E^{s_i}} \Omega'(s_i, a; \pi_E) + \Omega(\pi_E^{s_i}) - \Omega(\pi^{s_i}) \right\} \right]$$

$$\stackrel{(iv)}{=} \mathbb{E}_\pi \left[ \sum_{i=0}^\infty \gamma^i \left\{ \langle \nabla \Omega(\pi_E^{s_i}), \pi^{s_i} \rangle_{\mathcal{A}} - \langle \nabla \Omega(\pi_E^{s_i}), \pi_E^{s_i} \rangle_{\mathcal{A}} + \Omega(\pi_E^{s_i}) - \Omega(\pi^{s_i}) \right\} \right]$$

$$= \mathbb{E}_\pi \left[ \sum_{i=0}^\infty \gamma^i \left\{ \Omega(\pi_E^{s_i}) - \Omega(\pi^{s_i}) + \langle \nabla \Omega(\pi_E^{s_i}), \pi^{s_i} \rangle_{\mathcal{A}} - \langle \nabla \Omega(\pi_E^{s_i}), \pi_E^{s_i} \rangle_{\mathcal{A}} \right\} \right]$$

$$= -\mathbb{E}_\pi \left[ \sum_{i=0}^\infty \gamma^i \left\{ \Omega(\pi^{s_i}) - \Omega(\pi_E^{s_i}) - \langle \nabla \Omega(\pi_E^{s_i}), \pi^{s_i} \rangle_{\mathcal{A}} + \langle \nabla \Omega(\pi_E^{s_i}), \pi_E^{s_i} \rangle_{\mathcal{A}} \right\} \right]$$

$$= -\mathbb{E}_\pi \left[ \sum_{i=0}^\infty \gamma^i \left\{ \Omega(\pi^{s_i}) - \Omega(\pi_E^{s_i}) - \langle \nabla \Omega(\pi_E^{s_i}), \pi^{s_i} - \pi_E^{s_i} \rangle_{\mathcal{A}} \right\} \right] \stackrel{(v)}{=} -\mathbb{E}_\pi \left[ \sum_{i=0}^\infty \gamma^i D_\Omega^{\mathcal{A}}(\pi^{s_i} || \pi_E^{s_i})) \right],$$

$$\tag{16}$$

where (i) follows from the assumption $r(s, a) = t(s, a; \pi_E)$ in **Lemma 1**, (ii) follows from the definition of $t(s, a; \pi)$ in Eq.(7), (iii) follows since taking the inner expectation first does not change the overall expectation, (iv) follows from the definition of $\Omega'$ in **Lemma 1** and $\sum_{a \in \mathcal{A}} p(a)[\nabla\Omega(p)](a) = \langle \nabla\Omega(p), p \rangle_{\mathcal{A}}$, and (v) follows from the definition of Bregman divergence, i.e., $D_{\Omega}^{\mathcal{A}}(p_1 || p_2) = \Omega(p_1) - \Omega(p_2) - \langle \nabla\Omega(p_2), p_1 - p_2 \rangle_{\mathcal{A}}$. Due to the non-negativity of $D_{\Omega}^{\mathcal{A}}$, Eq.(16) is less than or equal to zero which can be achieved when $\pi = \pi_E$.

## C  PROOF OF **COROLLARY 1**

Let $a \in \{1, ..., |\mathcal{A}|\}$ and $\pi_a = \pi(a)$ for simplicity. For

$$\Omega(\pi) = -\lambda \mathbb{E}_{a \sim \pi} \phi(\pi_a) = -\lambda \sum_{a \in \mathcal{A}} \pi_a \phi(\pi_a) = -\lambda \sum_{a \in \mathcal{A}} f_\phi(\pi_a)$$

with $f_\phi(x) = x\phi(x)$, we have

$$\nabla\Omega(\pi) = -\lambda \nabla_{\pi_1, ..., \pi_{|\mathcal{A}|}} \sum_{a \in \mathcal{A}} f_\phi(\pi_a) = -\lambda [f'_\phi(\pi_1), ..., f'_\phi(\pi_{|\mathcal{A}|})]^T$$

for $f'_\phi(x) = \frac{\partial}{\partial x}(x\phi(x))$. Therefore, for $\pi^s = \pi(\cdot|s)$ we have

$$
\begin{aligned}
t(s, a; \pi) &= [\nabla\Omega(\pi^s)](a) - \langle \nabla\Omega(\pi^s), \pi^s \rangle_{\mathcal{A}} + \Omega(\pi^s) \\
&= -\lambda f'_\phi(\pi_a^s) - \left( \sum_{a' \in \mathcal{A}} \pi_{a'}^s(-\lambda f'_\phi(\pi_{a'}^s)) \right) + \left( -\lambda \sum_{a' \in \mathcal{A}} \pi_{a'}^s \phi(\pi_{a'}^s) \right) \\
&= -\lambda \left\{ f'_\phi(\pi_a^s) - \left( \sum_{a' \in \mathcal{A}} \pi_{a'}^s f'_\phi(\pi_{a'}^s) \right) + \left( \sum_{a' \in \mathcal{A}} \pi_{a'}^s \phi(\pi_{a'}^s) \right) \right\} \\
&= -\lambda \left\{ f'_\phi(\pi_a^s) - \sum_{a' \in \mathcal{A}} \pi_{a'}^s \left( f'_\phi(\pi_{a'}^s) - \phi(\pi_{a'}^s) \right) \right\} \\
&= -\lambda \left\{ f'_\phi(\pi_a^s) - \mathbb{E}_{a' \sim \pi^s} \left[ f'_\phi(\pi_{a'}^s) - \phi(\pi_{a'}^s) \right] \right\}.
\end{aligned}
$$

## D  PROOF OF OPTIMAL REWARDS ON CONTINUOUS CONTROLS

Note that for two continuous distributions $\mathbb{P}_1$ and $\mathbb{P}_2$ having probability density functions $p_1(x)$ and $p_2(x)$, respectively, the Bregman divergence can be defined as (Guo et al., 2017; Jones & Byrne, 1990)

$$D_\omega^{\mathcal{X}}(\mathbb{P}_1 || \mathbb{P}_2) := \int_{\mathcal{X}} \{ \omega(p_1(x)) - \omega(p_2(x)) - \omega'(p_2(x))(p_1(x) - p_2(x)) \} \, dx, \qquad (17)$$

where $\omega'(x) := \frac{\partial}{\partial x}\omega(x)$ and the divergence is measured point-wisely on $x \in \mathcal{X}$. Let us assume

$$\Omega(\pi) = \int_{\mathcal{A}} \omega(\pi(a)) da \qquad (18)$$

for a probability density function $\pi$ on $\mathcal{A}$ and define

$$t(s, a; \pi) := \omega'(\pi^s(a)) - \int_{\mathcal{A}} [\pi^s(a')\omega'(\pi^s(a')) - \omega(\pi^s(a'))] \, da'. \qquad (19)$$

for $\pi^s = \pi(\cdot|s)$. For $r(s,a) = t(s,a;\pi_E)$, the RL objective in Eq.(1) satisfies

$$
\mathbb{E}_\pi \left[ \sum_{i=0}^{\infty} \gamma^i \left\{ r(s_i, a_i) - \Omega(\pi^{s_i}) \right\} \right] \stackrel{(i)}{=} \mathbb{E}_\pi \left[ \sum_{i=0}^{\infty} \gamma^i \left\{ t(s_i, a_i; \pi_E) - \Omega(\pi^{s_i}) \right\} \right]
$$

$$
\stackrel{(ii)}{=} \mathbb{E}_\pi \left[ \sum_{i=0}^{\infty} \gamma^i \left\{ \omega'(\pi_E^{s_i}(a_i)) - \int_{\mathcal{A}} \left[ \pi_E^{s_i}(a)\omega'(\pi_E^{s_i}(a)) - \omega(\pi_E^{s_i}(a)) \right] da - \int_{\mathcal{A}} \omega(\pi^{s_i}(a)) da \right\} \right]
$$

$$
= \mathbb{E}_\pi \left[ \sum_{i=0}^{\infty} \gamma^i \left\{ \int_{\mathcal{A}} \pi^{s_i}(a)\omega'(\pi_E^{s_i}(a)) da - \int_{\mathcal{A}} \left[ \pi_E^{s_i}(a)\omega'(\pi_E^{s_i}(a)) - \omega(\pi_E^{s_i}(a)) + \omega(\pi^{s_i}(a)) \right] da \right\} \right]
$$

$$
= \mathbb{E}_\pi \left[ \sum_{i=0}^{\infty} \gamma^i \int_{\mathcal{A}} \left\{ \pi^{s_i}(a)\omega'(\pi_E^{s_i}(a)) - \left[ \pi_E^{s_i}(a)\omega'(\pi_E^{s_i}(a)) - \omega(\pi_E^{s_i}(a)) + \omega(\pi^{s_i}(a)) \right] \right\} da \right]
$$

$$
= \mathbb{E}_\pi \left[ \sum_{i=0}^{\infty} \gamma^i \int_{\mathcal{A}} \left\{ \omega(\pi_E^{s_i}(a)) - \omega(\pi^{s_i}(a)) + \pi^{s_i}(a)\omega'(\pi_E^{s_i}(a)) - \pi_E^{s_i}(a)\omega'(\pi_E^{s_i}(a)) \right\} da \right]
$$

$$
= -\mathbb{E}_\pi \left[ \sum_{i=0}^{\infty} \gamma^i \int_{\mathcal{A}} \left\{ \omega(\pi^{s_i}(a)) - \omega(\pi_E^{s_i}(a)) - \pi^{s_i}(a)\omega'(\pi_E^{s_i}(a)) + \pi_E^{s_i}(a)\omega'(\pi_E^{s_i}(a)) \right\} da \right]
$$

$$
= -\mathbb{E}_\pi \left[ \sum_{i=0}^{\infty} \gamma^i \int_{\mathcal{A}} \left\{ \omega(\pi^{s_i}(a)) - \omega(\pi_E^{s_i}(a)) - \omega'(\pi_E^{s_i}(a)) \left( \pi^{s_i}(a) - \pi_E^{s_i}(a) \right) \right\} da \right]
$$

$$
\stackrel{(iii)}{=} -\mathbb{E}_\pi \left[ \sum_{i=0}^{\infty} \gamma^i D_\omega^{\mathcal{A}}(\pi^{s_i} || \pi_E^{s_i}) \right], \tag{20}
$$

where (i) follows from $r(s,a) = t(s,a;\pi_E)$, (ii) follows from Eq.(18) and Eq.(19), and (iii) follows from the definition of Bregman divergence in Eq.(17). Due to the non-negativity of $D_\omega$, Eq.(24) is less than or equal to zero which can be achieved when $\pi = \pi_E$. Also, $\pi = \pi_E$ is a unique solution since Eq.(1) has a unique solution for arbitrary reward functions.

# E  PROOF OF LEMMA 2

Since **Lemma 2** was mentioned but not proved in Geist et al. (2019), we derive the rigorous proof for **Lemma 2** in this subsection. Note that we follow the proof idea in Ng et al. (1999).

First, let us assume an MDP $\mathcal{M}^r$ with a reward $r$ and corresponding optimal policy $\pi^*$. From **Definition 1**, the optimal state-value function $V^{*,r}$ and its corresponding state-action value function $Q^{*,r}(s,a) := r(s,a) + \gamma\mathbb{E}_{s'\sim P(\cdot|s,a)}[V^{*,r}(s')]$ should satisfy the regularized Bellman optimality equation

$$
\begin{aligned}
V^{*,r}(s) = T^{*,r}V^{*,r}(s) &= \Omega^*(Q^{*,r}(s,a)) \\
&= \max_{\pi(\cdot|s)\in\Delta^{\mathcal{A}}} \langle \pi(\cdot|s), Q^{*,r}(s,a) \rangle_{\mathcal{A}} - \Omega(\pi(\cdot|s)) \\
&= \max_{\pi(\cdot|s)\in\Delta^{\mathcal{A}}} \langle \pi(\cdot|s), r(s,a) + \gamma\mathbb{E}_{s'\sim P(\cdot|s,a)}[V^{*,r}(s')] \rangle_{\mathcal{A}} - \Omega(\pi(\cdot|s)),
\end{aligned} \tag{21}
$$

where we explicitize the dependencies on $r$. Also, the optimal policy $\pi^*$ is a unique maximizer for the above maximization.

Now, let us consider the shaped rewards $r(s,a) + \Phi(s') - \Phi(s)$ and $r(s,a) + \mathbb{E}_{s'\sim P(\cdot|s,a)}\Phi(s') - \Phi(s)$. Please note that for both rewards, the expectation over $s'$ for given $s, a$ is equal to

$$
\tilde{r}(s,a) = r(s,a) + \mathbb{E}_{s'\sim P(\cdot|s,a)}\Phi(s') - \Phi(s),
$$

and thus, it is sufficient to only consider the optimality for $\tilde{r}$. By subtracting $\Phi(s)$ from the regularized optimality Bellman equation for $r$ in Eq.(21), we have

$$V^{*,r}(s) - \Phi(s)$$
$$= \max_{\pi(\cdot|s) \in \Delta^{\mathcal{A}}} \langle \pi(\cdot|s), r(s,a) + \gamma \mathbb{E}_{s' \sim P(\cdot|s,a)} \Phi(s') - \Phi(s) + \gamma \mathbb{E}_{s' \sim P(\cdot|s,a)} [V^{*,r}(s') - \Phi(s')] \rangle_{\mathcal{A}}$$
$$\qquad - \Omega(\pi(\cdot|s))$$
$$= \max_{\pi(\cdot|s) \in \Delta^{\mathcal{A}}} \langle \pi(\cdot|s), \tilde{r}(s,a) + \gamma \mathbb{E}_{s' \sim P(\cdot|s,a)} [V^{*,r}(s') - \Phi(s')] \rangle_{\mathcal{A}} - \Omega(\pi(\cdot|s))$$
$$= [T^{*,\tilde{r}}(V^{*,r} - \Phi)](s).$$

That is, $V^{*,r} - \Phi$ is the fixed point of the regularized Bellman optimality operator $T^{*,\tilde{r}}$ associated with the shaped reward $\tilde{r}$. Also, a maximizer for the above maximization is $\pi^*$ since subtracting $\Phi(s)$ from Eq.(21) does not change the unique maximizer $\pi^*$.

## F    COMPARISON BETWEEN OUR SOLUTION AND EXISTING SOLUTION IN GEIST ET AL. (2019)

### F.1    ISSUES WITH THE SOLUTIONS IN GEIST ET AL. (2019)

In **Proposition 5** of Geist et al. (2019), a solution of regularized IRL is given, and we rewrite the relevant theorem in this subsection. Let us consider a regularized IRL problem, where $\pi_{E} \in \Delta_{\mathcal{S}}^{\mathcal{A}}$ is an expert policy. Assuming that both the model (dynamics, discount factor and regularizer) and the expert policy are known, Geist et al. (2019) proposed a solution of regularized IRL as follows:

**Lemma 4** (A solution of regularized IRL from Geist et al. (2019)). *Let $Q_E \in \mathbb{R}^{\mathcal{S} \times \mathcal{A}}$ be a function such that $\pi_E(\cdot|s) = \nabla\Omega^*(Q_E(s,\cdot)), s \in \mathcal{S}$. Also, define*

$$r_E(s,a) := Q_E(s,a) - \gamma\mathbb{E}_{s' \sim P(\cdot|s,a)}[\Omega^*(Q_E(s',\cdot))]$$
$$= Q_E(s,a) - \gamma\mathbb{E}_{s' \sim P(\cdot|s,a)}[\langle \pi_E(\cdot|s'), Q_E(s',\cdot) \rangle_{\mathcal{A}} - \Omega(\pi_E(\cdot|s'))].$$

*Then, in the $\Omega$-regularized MDP with the reward $r_E$, $\pi_E$ is an optimal policy.*

*Proof.* Although the brief idea of the proof is given in Geist et al. (2019), we rewrite the proof in a more rigorous way as follows. Let us define $V_E(s) = \langle \pi_E(\cdot|s), Q_E(s,\cdot) \rangle_{\mathcal{A}} - \Omega(\pi_E(\cdot|s))$. Then, $r_E(s,a) = Q_E(s,a) - \gamma\mathbb{E}_{s' \sim P(\cdot|s,a)}[V_E(s')]$, and thus, $Q_E(s,a) = r_E(s,a) + \gamma\mathbb{E}_{s' \sim P(\cdot|s,a)}[V_E(s')]$. By using this and regularized Bellman optimality operator, we have

$$[T^*V_E](s) \overset{(i)}{=} \Omega^*(Q_E(s,\cdot)) \overset{(ii)}{=} \max_{\pi(\cdot|s) \in \Delta^{\mathcal{A}}} \langle \pi(\cdot|s), Q_E(s,\cdot) \rangle_{\mathcal{A}} - \Omega(\pi(\cdot|s))$$

$$\overset{(iii)}{=} \langle \pi_E(\cdot|s), Q_E(s,\cdot) \rangle_{\mathcal{A}} - \Omega(\pi_E(\cdot|s)) = V_E(s),$$

where (i) and (ii) follow from **Definition 1**, and (iii) follows since $\pi_E$ is a unique maximizer. Thus, $V_E$ is the fixed point of $T^*$, and $\pi_E(\cdot|s) = \nabla\Omega^*(Q_E(s,\cdot)), s \in \mathcal{S}$, becomes an optimal policy.    □

For example, when negative Shannon entropy is used as a regularizer, we can get $r_E(s,a) = \log \pi_E(a|s)$ by setting $Q_E(s,a) = \log \pi_E(a|s)$. However, a solution proposed in **Lemma 4** has two crucial issues:
**Issue 1.** It requires the knowledge on the model dynamics, which is generally intractable.
**Issue 2.** We need to figure out $Q_E$ that satisfies $\pi_E(\cdot|s) = \nabla\Omega^*(Q_E(s,\cdot))$, which comes from the relationship between the optimal value function and optimal policy (Geist et al., 2019).

In the following subsection, we show how our solution in **Lemma 1** is related to the solution from Geist et al. (2019) in **Lemma 4**.

### F.2    RELATION BETWEEN THE SOLUTION OF GEIST ET AL. (2019) AND OUR TRACTABLE SOLUTION

Let us consider the expert policy $\pi_E$ and functions $Q_E$ and $r_E$ satisfying the conditions in **Lemma 4**. From **Lemma 2**, a regularized MDP with the shaped reward

$$\tilde{r}_E(s,a) := r_E(s,a) + \gamma\mathbb{E}_{s' \sim P(\cdot|s,a)}\Phi(s') - \Phi(s)$$

for $\Phi \in \mathbb{R}^{\mathcal{S}}$ has its optimal policy as $\pi_E$. Since $\Phi$ can be arbitrarily chosen, let us assume $\Phi(s) = \Omega^*(Q_E(s, \cdot))$. Then, we have

$$
\begin{aligned}
&\tilde{r}_E(s, a) \\
&= r_E(s, a) + \gamma \mathbb{E}_{s' \sim P(\cdot|s,a)} \Phi(s') - \Phi(s) \\
&= \left\{ Q_E(s, a) - \gamma \mathbb{E}_{s' \sim P(\cdot|s,a)}[\Omega^*(Q_E(s', \cdot))] \right\} + \gamma \mathbb{E}_{s' \sim P(\cdot|s,a)} \Omega^*(Q_E(s', \cdot)) - \Omega^*(Q_E(s, \cdot)) \\
&= Q_E(s, a) - \Omega^*(Q_E(s, \cdot)).
\end{aligned}
\tag{22}
$$

Note that the reward in $Eq.(22)$ does not require the knowledge on the model dynamics, which addresses **Issue 1** in Appendix F.1. Also, by using $V_E(s) = \Omega^*(Q_E(s, \cdot))$ in the proof of **Lemma 4**, the reward in $Eq.(22)$ can be written as

$$
\tilde{r}_E(s, a) = Q_E(s, a) - V_E(s),
$$

which means $\tilde{r}_E(s, a)$ is *an advantage function* for the optimal policy $\pi_E$ in the $\Omega$-regularized MDP.

However, we still have **Issue 2** in Appendix F.1 since $\Omega^*$ in $Eq.(22)$ is generally intractable (Geist et al., 2019), which prevents us from finding the appropriate $Q_E(s, a)$. Interestingly, we show that for all $s \in S$ and $Q_E(s, \cdot) = \nabla \Omega(\pi_E(\cdot|s))$,

$$
\begin{aligned}
\nabla \Omega^*(Q_E(s, \cdot)) &= \underset{\pi(\cdot|s) \in \Delta^{\mathcal{A}}}{\arg\max} \; \langle \pi(\cdot|s), Q_E(s, \cdot) \rangle_{\mathcal{A}} - \Omega(\pi(\cdot|s)) \\
&= \underset{\pi(\cdot|s) \in \Delta^{\mathcal{A}}}{\arg\max} \; \langle \pi(\cdot|s), \nabla \Omega(\pi_E(\cdot|s)) \rangle_{\mathcal{A}} - \Omega(\pi(\cdot|s)) \\
&= \underset{\pi(\cdot|s) \in \Delta^{\mathcal{A}}}{\arg\min} \; \Omega(\pi(\cdot|s)) - \langle \nabla \Omega(\pi_E(\cdot|s)), \pi(\cdot|s) \rangle_{\mathcal{A}} \\
&= \underset{\pi(\cdot|s) \in \Delta^{\mathcal{A}}}{\arg\min} \; \Omega(\pi(\cdot|s)) - \Omega(\pi_E(\cdot|s)) - \langle \nabla \Omega(\pi_E(\cdot|s)), \pi(\cdot|s) - \pi_E(\cdot|s) \rangle_{\mathcal{A}} \\
&= \underset{\pi(\cdot|s) \in \Delta^{\mathcal{A}}}{\arg\min} \; D_\Omega^{\mathcal{A}}(\pi(\cdot|s)||\pi_E(\cdot|s)) = \pi_E(\cdot|s),
\end{aligned}
$$

where the last equality holds since the Bregman divergence $D_\Omega^{\mathcal{A}}(\pi(\cdot|s)||\pi_E(\cdot|s))$ is greater than or equal to zero and its lowest value is achieved when $\pi(\cdot|s) = \pi_E(\cdot|s)$. This means that when $Q_E(s, \cdot) = \nabla \Omega(\pi_E(\cdot|s))$ is used, the condition $\pi_E(\cdot|s) = \nabla \Omega^*(Q_E(s, \cdot))$ in **Lemma 4** is satisfied *without* knowing the tractable form of $\Omega^*$ or $\nabla \Omega^*$. Thus, **Issue 2** in Appendix F.1 is addressed; we instead require the knowledge on the gradient $\nabla \Omega$ of the policy regularizer $\Omega$, which is practically more tractable. Finally, by substituting $Q_E(s, \cdot) = \Omega'(s, \cdot; \pi_E)$ for $\Omega'(s, \cdot; \pi) := \nabla \Omega(\pi(\cdot|s)), s \in \mathcal{S}$, to Eq.(22), we have

$$
\begin{aligned}
\tilde{r}_E(s, a) &= \Omega'(s, a; \pi_E) - \Omega^*(\Omega'(s, \cdot; \pi_E)) \\
&= \Omega'(s, a; \pi_E) - \left\{ \langle \pi_E(\cdot|s), \Omega'(s, \cdot; \pi_E) \rangle_{\mathcal{A}} - \Omega(\pi_E(\cdot|s)) \right\} \\
&= \Omega'(s, a; \pi_E) - \mathbb{E}_{a' \sim \pi_E(\cdot|s)}[\Omega'(s, a'; \pi_E)] + \Omega(\pi_E(\cdot|s)) \\
&= t(s, a; \pi_E),
\end{aligned}
$$

where $t(s, a; \pi_E)$ is our proposed solution in **Lemma 1**.

# G  PROOF OF **LEMMA 3**

**RL objective in Regularized MDPs w.r.t. normalized visitation distributions.** For a reward function $r \in \mathbb{R}^{\mathcal{S} \times \mathcal{A}}$ and a strongly convex function $\Omega : \Delta^{\mathcal{A}} \to \mathbb{R}$, the RL objective $J_\Omega(r, \pi)$ in Eq.(1) is equivalent to

$$
\underset{\pi}{\arg\max} \; \bar{J}_{\bar{\Omega}}(r, d_\pi) := \langle r, d_\pi \rangle_{\mathcal{S} \times \mathcal{A}} - \bar{\Omega}(d_\pi),
\tag{23}
$$

where for a set $\mathcal{D}$ of *normalized* visitation distributions (Syed et al., 2008)

$$
\mathcal{D} := \left\{ d \in \mathbb{R}^{\mathcal{S} \times \mathcal{A}} : \sum_{a'} d(s', a') = (1 - \gamma)P_0(s') + \gamma \sum_{s,a} P(s'|s, a)d(s, a), \forall s' \in \mathcal{S} \right\},
$$

we define $\bar{\Omega}(d) := \mathbb{E}_{(s,a)\sim d}[\Omega(\bar{\pi}_d(\cdot|s))]$ and $\bar{\pi}_d(\cdot|s) := \frac{d(s,\cdot)}{\sum_{a'} d(s,a')} \in \Delta_{\mathcal{S}}^{\mathcal{A}}$ for $d \in \mathcal{D}$ and use $\bar{\pi}_{d_\pi}(\cdot|s) = \pi(\cdot|s)$ for all $s \in \mathcal{S}$. For $\bar{\Omega} : \mathcal{D} \to \mathbb{R}$, its convex conjugate $\bar{\Omega}^*$ is

$$
\begin{aligned}
\bar{\Omega}^*(r) :&= \max_{d\in\mathcal{D}} \bar{J}_{\bar{\Omega}}(r,d) \\
&= \max_{d\in\mathcal{D}} \langle r, d\rangle_{\mathcal{S}\times\mathcal{A}} - \bar{\Omega}(d) \\
&\overset{(i)}{=} \max_{\pi\in\Delta_{\mathcal{S}}^{\mathcal{A}}} \langle r, d_\pi\rangle_{\mathcal{S}\times\mathcal{A}} - \bar{\Omega}(d_\pi) \\
&= \max_{\pi\in\Delta_{\mathcal{S}}^{\mathcal{A}}} \sum_{s,a} d_\pi(s,a)\left[r(s,a) - \Omega(\pi(a|s))\right] \\
&= (1-\gamma)\cdot \max_{\pi\in\Delta_{\mathcal{S}}^{\mathcal{A}}} J_\Omega(r,\pi),
\end{aligned}
\tag{24}
$$

where (i) follows from using the one-to-one correspondence between policies and visitation distributions (Syed et al., 2008; Ho & Ermon, 2016). Note that Eq.(24) is equal to the optimal discounted average return in regularized MDPs.

**IRL objective in regularized MDPs w.r.t. normalized visitation distributions.** By using the RL objective in Eq.(23), we can rewrite the IRL objective in Eq.(6) w.r.t. the normalized visitation distributions as the maximization of the following objective over $r \in \mathbb{R}^{\mathcal{S}\times\mathcal{A}}$:

$$
\begin{aligned}
&(1-\gamma)\cdot\left\{ J_\Omega(r,\pi_E) - \max_{\pi\in\Delta_{\mathcal{S}}^{\mathcal{A}}} J_\Omega(r,\pi) \right\} \\
&= \bar{J}_{\bar{\Omega}}(r, d_{\pi_E}) - \max_{d\in\mathcal{D}} \bar{J}_{\bar{\Omega}}(r,d) \\
&= \min_{d\in\mathcal{D}}\left\{ \bar{J}_{\bar{\Omega}}(r, d_{\pi_E}) - \bar{J}_{\bar{\Omega}}(r,d) \right\} \\
&= \min_{d\in\mathcal{D}}\left\{ \left(\langle r, d_{\pi_E}\rangle_{\mathcal{S}\times\mathcal{A}} - \bar{\Omega}(d_{\pi_E})\right) - \left(\langle r, d\rangle_{\mathcal{S}\times\mathcal{A}} - \bar{\Omega}(d)\right) \right\} \\
&= \min_{d\in\mathcal{D}}\left\{ \bar{\Omega}(d) - \bar{\Omega}(d_{\pi_E}) - \langle r, d - d_{\pi_E}\rangle_{\mathcal{S}\times\mathcal{A}} \right\}.
\end{aligned}
\tag{25}
$$

Note that if $\nabla\bar{\Omega}(d)$ is well-defined and $r = \nabla\bar{\Omega}(d_{\pi_E})$ for any strictly convex $\bar{\Omega}$, Eq.(25) is equal to

$$
\min_{d\in\mathcal{D}}\left\{ \bar{\Omega}(d) - \bar{\Omega}(d_{\pi_E}) - \langle \nabla\bar{\Omega}(d_{\pi_E}), d - d_{\pi_E}\rangle_{\mathcal{S}\times\mathcal{A}} \right\} = \min_{d\in\mathcal{D}} D_{\bar{\Omega}}^{\mathcal{S}\times\mathcal{A}}(d||d_{\pi_E}),
$$

where the equality comes from the definition of Bregman divergence.

**Proof of** $t(s,a;\pi_d) = \nabla[\bar{\Omega}(d)](s,a)$**.** For simpler notation, we use matrix-vector notation for the proof when discrete state and action spaces $\mathcal{S} = \{1,...,|\mathcal{S}|\}$ and $\mathcal{A} = \{1,...,|\mathcal{A}|\}$ are considered. For a normalized visitation distribution $d \in \mathcal{D}$, let us define

$$
\begin{aligned}
&d_a^s := d(s,a), s \in \mathcal{S}, a \in \mathcal{A}, \\
&\boldsymbol{d}^s := [d_1^s, ..., d_{|\mathcal{A}|}^s]^T \in \mathbb{R}^{\mathcal{A}}, s \in \mathcal{S}, \\
&\boldsymbol{D} := [\boldsymbol{d}^1, ..., \boldsymbol{d}^{|\mathcal{S}|}]^T = \begin{bmatrix} d_1^1 & \cdots & d_{|\mathcal{A}|}^1 \\ \vdots & \ddots & \vdots \\ d_1^{|\mathcal{S}|} & \cdots & d_{|\mathcal{A}|}^{|\mathcal{S}|} \end{bmatrix} \in \mathbb{R}^{\mathcal{S}\times\mathcal{A}}, \\
&\bar{\boldsymbol{\pi}}(\boldsymbol{x}) := \frac{\boldsymbol{x}}{\boldsymbol{1}_{\mathcal{A}}^T\boldsymbol{x}} = \frac{1}{\sum_{a\in\mathcal{A}} x_a}\left[x_1,...,x_{|\mathcal{A}|}\right]^T \in \mathbb{R}^{\mathcal{A}}, \boldsymbol{x} := \left[x_1,...,x_{|\mathcal{A}|}\right]^T \in \mathbb{R}^{\mathcal{A}},
\end{aligned}
$$

where $\boldsymbol{1}_{\mathcal{A}} = [1,...,1]^T \in \mathbb{R}^{\mathcal{A}}$ is an $|\mathcal{A}|$-dimensional all-one vector. By using these notations, the original $\bar{\Omega}$ can be rewritten as

$$
\bar{\Omega}(\boldsymbol{D}) = \sum_{s,a} d_a^s \Omega(\bar{\boldsymbol{\pi}}(\boldsymbol{d}^s)) = \sum_{s\in\mathcal{S}} \boldsymbol{1}_{\mathcal{A}}^T\boldsymbol{d}^s \Omega(\bar{\boldsymbol{\pi}}(\boldsymbol{d}^s)).
$$

The gradient of $\bar\Omega$ w.r.t. $\boldsymbol{D}$ (using denominator-layout notation) is

$$\nabla_{\boldsymbol{D}}\bar\Omega(\boldsymbol{D}) = \left[\frac{\partial\bar\Omega(\boldsymbol{D})}{\partial\boldsymbol{d}^1}, ..., \frac{\partial\bar\Omega(\boldsymbol{D})}{\partial\boldsymbol{d}^{|\mathcal{S}|}}\right]^T \in \mathbb{R}^{\mathcal{S}\times\mathcal{A}},$$

where each element of $\nabla_{\boldsymbol{D}}\bar\Omega(\boldsymbol{D})$ satisfies

$$\begin{aligned}
\frac{\partial\bar\Omega(\boldsymbol{D})}{\partial\boldsymbol{d}^s} &= \left[\frac{\partial\bar\Omega(\boldsymbol{D})}{\partial d_1^s}, ..., \frac{\partial\bar\Omega(\boldsymbol{D})}{\partial d_{|\mathcal{A}|}^s}\right]^T \\
&= \frac{\partial}{\partial\boldsymbol{d}^s}\left\{\sum_{s\in\mathcal{S}} \boldsymbol{1}_{\mathcal{A}}^T \boldsymbol{d}^s \Omega(\bar\pi(\boldsymbol{d}^s))\right\} \\
&= \Omega(\bar\pi(\boldsymbol{d}^s))\boldsymbol{1}_{\mathcal{A}} + \boldsymbol{1}_{\mathcal{A}}^T\boldsymbol{d}^s \frac{\partial\Omega(\bar\pi(\boldsymbol{d}^s))}{\partial\boldsymbol{d}^s} \\
&= \Omega(\bar\pi(\boldsymbol{d}^s))\boldsymbol{1}_{\mathcal{A}} + \boldsymbol{1}_{\mathcal{A}}^T\boldsymbol{d}^s \frac{\partial\bar\pi(\boldsymbol{d}^s)}{\partial\boldsymbol{d}^s}\frac{\partial\Omega(\bar\pi(\boldsymbol{d}^s))}{\partial\bar\pi(\boldsymbol{d}^s)}.
\end{aligned} \tag{26}$$

for

$$\frac{\partial\bar\pi(\boldsymbol{d}^s)}{\partial\boldsymbol{d}^s} = \left[\frac{\partial\bar\pi_1(\boldsymbol{d}^s)}{\partial\boldsymbol{d}^s}, ..., \frac{\partial\bar\pi_{|\mathcal{A}|}(\boldsymbol{d}^s)}{\partial\boldsymbol{d}^s}\right],$$

$$\frac{\partial\bar\pi_a(\boldsymbol{d}^s)}{\partial\boldsymbol{d}^s} = \frac{\partial}{\partial\boldsymbol{d}^s}\left[\frac{d_a^s}{\boldsymbol{1}_{\mathcal{A}}^T\boldsymbol{d}^s}\right] = \frac{\partial d_a^s}{\partial\boldsymbol{d}^s}(\boldsymbol{1}_{\mathcal{A}}^T\boldsymbol{d}^s)^{-1} + d_a^s\frac{\partial(\boldsymbol{1}_{\mathcal{A}}^T\boldsymbol{d}^s)^{-1}}{\partial\boldsymbol{d}^s}.$$

Note that each element of $\frac{\partial\bar\pi_a(\boldsymbol{d}^s)}{\partial\boldsymbol{d}^s}$ satisfies

$$\begin{aligned}
\frac{\partial\bar\pi_a(\boldsymbol{d}^s)}{\partial d_{a'}^s} &= \frac{\partial d_a^s}{\partial d_{a'}^s}(\boldsymbol{1}_{\mathcal{A}}^T\boldsymbol{d}^s)^{-1} + d_a^s\frac{\partial(\boldsymbol{1}_{\mathcal{A}}^T\boldsymbol{d}^s)^{-1}}{\partial d_{a'}^s} \\
&= \mathbb{I}\{a = a'\}(\boldsymbol{1}_{\mathcal{A}}^T\boldsymbol{d}^s)^{-1} - d_a^s(\boldsymbol{1}_{\mathcal{A}}^T\boldsymbol{d}^s)^{-2} \\
&= \mathbb{I}\{a = a'\}(\boldsymbol{1}_{\mathcal{A}}^T\boldsymbol{d}^s)^{-1} - \bar\pi_a(\boldsymbol{d}^s)(\boldsymbol{1}_{\mathcal{A}}^T\boldsymbol{d}^s)^{-1} \\
&= (\boldsymbol{1}_{\mathcal{A}}^T\boldsymbol{d}^s)^{-1}\left[\mathbb{I}\{a = a'\} - \bar\pi_a(\boldsymbol{d}^s)\right],
\end{aligned}$$

and thus,

$$\frac{\partial\bar\pi(\boldsymbol{d}^s)}{\partial\boldsymbol{d}^s} = (\boldsymbol{1}_{\mathcal{A}}^T\boldsymbol{d}^s)^{-1}\left\{\boldsymbol{I}_{\mathcal{A}\times\mathcal{A}} - \boldsymbol{1}_{\mathcal{A}}[\bar\pi(\boldsymbol{d}^s)]^T\right\}. \tag{27}$$

By substituting Eq.(27) into Eq.(26), we have

$$\begin{aligned}
\frac{\partial\bar\Omega(\boldsymbol{D})}{\partial\boldsymbol{d}^s} &= \Omega(\bar\pi(\boldsymbol{d}^s))\boldsymbol{1}_{\mathcal{A}} + \boldsymbol{1}_{\mathcal{A}}^T\boldsymbol{d}^s\frac{\partial\bar\pi(\boldsymbol{d}^s)}{\partial\boldsymbol{d}^s}\frac{\partial\Omega(\bar\pi(\boldsymbol{d}^s))}{\partial\bar\pi(\boldsymbol{d}^s)} \\
&= \Omega(\bar\pi(\boldsymbol{d}^s))\boldsymbol{1}_{\mathcal{A}} + \boldsymbol{1}_{\mathcal{A}}^T\boldsymbol{d}^s\left[(\boldsymbol{1}_{\mathcal{A}}^T\boldsymbol{d}^s)^{-1}\left\{\boldsymbol{I}_{\mathcal{A}\times\mathcal{A}} - \boldsymbol{1}_{\mathcal{A}}[\bar\pi(\boldsymbol{d}^s)]^T\right\}\right]\frac{\partial\Omega(\bar\pi(\boldsymbol{d}^s))}{\partial\bar\pi(\boldsymbol{d}^s)} \\
&= \Omega(\bar\pi(\boldsymbol{d}^s))\boldsymbol{1}_{\mathcal{A}} + \left\{\boldsymbol{I}_{\mathcal{A}\times\mathcal{A}} - \boldsymbol{1}_{\mathcal{A}}[\bar\pi(\boldsymbol{d}^s)]^T\right\}\frac{\partial\Omega(\bar\pi(\boldsymbol{d}^s))}{\partial\bar\pi(\boldsymbol{d}^s)} \\
&= \Omega(\bar\pi(\boldsymbol{d}^s))\boldsymbol{1}_{\mathcal{A}} + \frac{\partial\Omega(\bar\pi(\boldsymbol{d}^s))}{\partial\bar\pi(\boldsymbol{d}^s)} - [\bar\pi(\boldsymbol{d}^s)]^T\frac{\partial\Omega(\bar\pi(\boldsymbol{d}^s))}{\partial\bar\pi(\boldsymbol{d}^s)}\boldsymbol{1}_{\mathcal{A}} \\
&= \frac{\partial\Omega(\bar\pi(\boldsymbol{d}^s))}{\partial\bar\pi(\boldsymbol{d}^s)} - [\bar\pi(\boldsymbol{d}^s)]^T\frac{\partial\Omega(\bar\pi(\boldsymbol{d}^s))}{\partial\bar\pi(\boldsymbol{d}^s)}\boldsymbol{1}_{\mathcal{A}} + \Omega(\bar\pi(\boldsymbol{d}^s))\boldsymbol{1}_{\mathcal{A}}.
\end{aligned} \tag{28}$$

If we use the function notation, Eq.(28) can be written as

$$\begin{aligned}
\nabla[\bar\Omega(d)](s, a) &= \nabla\Omega(\bar\pi_d(\cdot|s))(a) - \mathbb{E}_{a'\sim\bar\pi_d(\cdot|s)}\left[\nabla\Omega(\bar\pi_d(\cdot|s))(a')\right] + \Omega(\bar\pi_d(\cdot|s)) \\
&= t(s, a; \bar\pi_d)
\end{aligned}$$

for $t$ of Eq.(7) in **Lemma 1**.

# H   DERIVATION OF BREGMAN-DIVERGENCE-BASED MEASURE IN CONTINUOUS CONTROLS

In Eq.(17), the Bregman divergence in the control task is defined as

$$D_\omega^{\mathcal{A}}(\mathbb{P}_1||\mathbb{P}_2) := \int_{\mathcal{X}} \{\omega(p_1(x)) - \omega(p_2(x)) - \omega'(p_2(x))(p_1(x) - p_2(x))\}\, dx. \tag{29}$$

Note that we consider $\Omega(p) = \int_{\mathcal{X}} \omega(p(x))dx = \int_{\mathcal{X}} [-f_\phi(p(x))]\, dx$ for $f_\phi(x) = x\phi(x)$, which makes Eq.(29) equal to

$$\int_{\mathcal{X}} \{-p_1(x)\phi(p_1(x)) + p_2(x)\phi(p_2(x)) + f_\phi'(p_2(x))(p_1(x) - p_2(x))\}\, dx$$

$$= \int_{\mathcal{X}} p_1(x) \{f_\phi'(p_2(x)) - \phi(p_1(x))\}\, dx - \int_{\mathcal{X}} p_2(x) \{f_\phi'(p_2(x)) - \phi(p_2(x))\}\, dx$$

$$= \mathbb{E}_{x\sim p_1} [f_\phi'(p_2(x)) - \phi(p_1(x))] - \mathbb{E}_{x\sim p_2} [f_\phi'(p_2(x)) - \phi(p_2(x))].$$

Thus, by considering a learning agent's policy $\pi^s = \pi(\cdot|s)$, expert policy $\pi_E^s = \pi_E(\cdot|s)$, and the objective in Eq.(8) characterized by the Bregman divergence, we can think of the following measure between expert and agent policies:

$$\mathbb{E}_{s\sim d_\pi} [D_\Omega^{\mathcal{A}}(\pi^s||\pi_E^s)]$$

$$= \mathbb{E}_{s\sim d_\pi} \left[ \mathbb{E}_{a\sim\pi^s} [f_\phi'(\pi_E^s(a)) - \phi(\pi^s(a))] - \mathbb{E}_{a\sim\pi_E^s} [f_\phi'(\pi_E^s(a)) - \phi(\pi_E^s(a))] \right]. \tag{30}$$

# I   TSALLIS ENTROPY AND ASSOCIATED BREGMAN DIVERGENCE AMONG MULTI-VARIATE GAUSSIAN DISTRIBUTIONS

Based on the derivation in Nielsen & Nock (2011), we derive the Tsallis entropy and associated Bremgan divergence as follows. We first consider the distributions in the exponential family

$$\exp\left(\langle\theta, t(x)\rangle - F(\theta) + k(x)\right). \tag{31}$$

Note that for

$$\theta = \begin{bmatrix} \Sigma^{-1}\mu \\ -\frac{1}{2}\Sigma^{-1} \end{bmatrix} = \begin{bmatrix} \theta_1 \\ \theta_2 \end{bmatrix},$$

$$t(x) = \begin{bmatrix} x \\ xx^T \end{bmatrix},$$

$$F(\theta) = -\frac{1}{4}\theta_1^T\theta_2^{-1}\theta_1 + \frac{1}{2}\log|-\pi\theta_2^{-1}| = \frac{1}{2}\mu^T\Sigma^{-1}\mu + \frac{1}{2}\log(2\pi)^d|\Sigma|,$$

$$k(x) = 0,$$

we can recover the multi-variate Gaussian distribution (Nielsen & Nock, 2011):

$$\exp(\langle\theta, t(x)\rangle - F(\theta) + k(x)) \tag{32}$$

$$= \exp\left(\mu^T\Sigma^{-1}x - \frac{1}{2}\text{tr}(\Sigma^{-1}xx^T) - \frac{1}{2}\mu^T\Sigma^{-1}\mu - \frac{1}{2}\log(2\pi)^d|\Sigma|\right) \tag{33}$$

$$= \frac{1}{(2\pi)^{d/2}|\Sigma|^{1/2}} \exp\left(\mu^T\Sigma^{-1}x - \frac{1}{2}x^T\Sigma^{-1}x - \frac{1}{2}\mu^T\Sigma^{-1}\mu\right) \tag{34}$$

$$= \frac{1}{(2\pi)^{d/2}|\Sigma|^{1/2}} \exp\left(\frac{1}{2}(x-\mu)^T\Sigma^{-1}(x-\mu)\right). \tag{35}$$

For two distributions with $k(x) = 0$,

$$\pi(x) = \exp(\langle\theta, t(x)\rangle - F(\theta)), \hat{\pi}(x) = \exp(\langle\hat{\theta}, t(x)\rangle - F(\hat{\theta}))$$

that share $t$, $F$, and $k$, it can be shown that

$$I(\pi, \hat{\pi}; \alpha, \beta) = \int \pi(x)^\alpha \hat{\pi}(x)^\beta dx$$

$$= \exp\left(F(\alpha\theta + \beta\hat{\theta}) - \alpha F(\theta) - \beta F(\hat{\theta})\right)$$

since

$$\int \pi(x)^\alpha \hat{\pi}(x)^\beta dx$$

$$= \int \exp\left(\alpha\langle\theta, t(x)\rangle - \alpha F(\theta) + \beta\langle\hat{\theta}, t(x)\rangle - \beta F(\hat{\theta})\right) dx$$

$$= \int \exp\left(\langle\alpha\theta + \beta\hat{\theta}, t(x)\rangle - F(\alpha\theta + \beta\hat{\theta})\right) \exp\left(F(\alpha\theta + \beta\hat{\theta}) - \alpha F(\theta) - \beta F(\hat{\theta})\right) dx$$

$$= \exp\left(F(\alpha\theta + \beta\hat{\theta}) - \alpha F(\theta) - \beta F(\hat{\theta})\right) \int \exp\left(\langle\alpha\theta + \beta\hat{\theta}, t(x)\rangle - F(\alpha\theta + \beta\hat{\theta})\right) dx$$

$$= \exp\left(F(\alpha\theta + \beta\hat{\theta}) - \alpha F(\theta) - \beta F(\hat{\theta})\right).$$

## I.1 TSALLIS ENTROPY

For $\phi(x) = \frac{k}{q-1}(1 - x^{q-1})$ and $k = 1$, the Tsallis entropy of $\pi$ can be written as

$$\mathcal{T}_q(\pi) := \mathbb{E}_{x\sim\pi}\phi(x) = \int \pi(x)\frac{1 - \pi(x)^{q-1}}{q-1}dx$$

$$= \frac{1 - \int \pi(x)^q dx}{q-1}$$

$$= \frac{1}{q-1}\left(1 - I(\pi, \pi; q, 0)\right) = \frac{1 - \exp\left(F(q\theta) - qF(\theta)\right)}{q-1}.$$

If $\pi$ is a multivariate Gaussian distribution, we have

$$F(q\theta) = \frac{q}{2}\mu^T\Sigma^{-1}\mu + \frac{1}{2}\log(2\pi)^d|\Sigma| - \frac{1}{2}\log q^d,$$

$$qF(\theta) = \frac{q}{2}\mu^T\Sigma^{-1}\mu + \frac{q}{2}\log(2\pi)^d|\Sigma|,$$

$$F(q\theta) - qF(\theta) = \frac{1-q}{2}\log(2\pi)^d|\Sigma| - \frac{1}{2}\log q^d$$

$$= (1-q)\left\{\frac{d}{2}\log 2\pi + \frac{1}{2}\log|\Sigma| - \frac{d\log q}{2(1-q)}\right\}.$$

For $\Sigma = \text{diag}\{\sigma_1^2, ..., \sigma_d^2\}$, we have

$$F(q\theta) - qF(\theta) = (1-q)\left\{\frac{d}{2}\log 2\pi + \frac{1}{2}\log|\Sigma| - \frac{d\log q}{2(1-q)}\right\}$$

$$= (1-q)\left\{\frac{d}{2}\log 2\pi + \frac{1}{2}\log\prod_{i=1}^d \sigma_i^2 - \frac{d\log q}{2(1-q)}\right\}$$

$$= (1-q)\sum_{i=1}^d\left\{\frac{\log 2\pi}{2} + \log\sigma_i - \frac{\log q}{2(1-q)}\right\}.$$

I.2   TRACTABLE FORM OF BASELINE

For $\phi(x) = \frac{k}{q-1}(1 - x^{q-1})$, we have

$$
\begin{aligned}
f'_\phi(x) &= \frac{k}{q-1}(1 - qx^{q-1}) \\
&= \frac{k}{q-1}(q - qx^{q-1} - (q-1)) \\
&= \frac{qk}{q-1}(1 - x^{q-1}) - k \\
&= q\phi(x) - k.
\end{aligned}
$$

Therefore, the baseline can be rewritten as

$$
\mathbb{E}_{x \sim \pi}[-f'_\phi(x) + \phi(x)] = \mathbb{E}_{x \sim \pi}[k - q\phi(x) + \phi(x)] = (1 - q)\mathcal{T}_q(\pi) + k.
$$

For a multivariate Gaussian distribution $\pi$, the tractable form of $\mathbb{E}_{x \sim \pi}[-f'_\phi(x) + \phi(x)]$ can be derived by using that of Tsallis entropy $\mathcal{T}_q(\pi)$ of $\pi$.

I.3   BREGMAN DIVERGENCE WITH TSALLIS ENTROPY REGULARIZATION

In Eq.(30), we consider the following form of the Bregman divergence:

$$
\int \pi(x)\{f'_\phi(\hat{\pi}(x)) - \phi(\pi(x))\}dx - \int \hat{\pi}(x)\{f'_\phi(\hat{\pi}(x)) - \phi(\hat{\pi}(x))\}dx.
$$

For $\phi(x) = \frac{k}{q-1}(1 - x^{q-1})$, $f'_\phi(x) = \frac{k}{q-1}(1 - qx^{q-1}) = q\phi(x) - k$, and $k = 1$, the above form is equal to

$$
\begin{aligned}
&\int \pi(x)\left[\frac{1 - q\hat{\pi}(x)^{q-1}}{q-1}\right]dx - \mathcal{T}_q(\pi) - (q-1)\mathcal{T}_q(\hat{\pi}) + 1 \\
&= \frac{1}{q-1} - \frac{q}{q-1}\int \pi(x)\hat{\pi}(x)^{q-1}dx - \mathcal{T}_q(\pi) - (q-1)\mathcal{T}_q(\hat{\pi}) + 1 \\
&= \frac{q}{q-1} - \frac{q}{q-1}\int \pi(x)\hat{\pi}(x)^{q-1}dx - \mathcal{T}_q(\pi) - (q-1)\mathcal{T}_q(\hat{\pi}).
\end{aligned}
$$

For multivariate Gaussians

$$
\begin{aligned}
\pi(x) &= \mathcal{N}(x; \mu, \Sigma), \mu = [\nu_1, ..., \nu_d]^T, \Sigma = \text{diag}(\sigma_1^2, ..., \sigma_d^2), \\
\hat{\pi}(x) &= \mathcal{N}(x; \hat{\mu}, \hat{\Sigma}), \hat{\mu} = [\hat{\nu}_1, ..., \hat{\nu}_d]^T, \hat{\Sigma} = \text{diag}(\hat{\sigma}_1^2, ..., \hat{\sigma}_d^2),
\end{aligned}
$$

we have

$$
\int \pi(x)\hat{\pi}(x)^{q-1}dx = I(\pi, \hat{\pi}; 1, q-1) = \exp\left(F(\theta') - F(\theta) - (q-1)F(\hat{\theta})\right),
$$

where

$$
\theta = \begin{bmatrix} \Sigma^{-1}\mu \\ -\frac{1}{2}\Sigma^{-1} \end{bmatrix},
$$

$$
\hat{\theta} = \begin{bmatrix} \hat{\Sigma}^{-1}\hat{\mu} \\ -\frac{1}{2}\hat{\Sigma}^{-1} \end{bmatrix},
$$

$$
\theta' = \theta + (q-1)\hat{\theta} = \begin{bmatrix} \Sigma^{-1}\mu + (q-1)\hat{\Sigma}^{-1}\hat{\mu} \\ -\frac{1}{2}(\Sigma^{-1} + (q-1)\hat{\Sigma}^{-1}) \end{bmatrix} = \begin{bmatrix} \theta'_1 \\ \theta'_2 \end{bmatrix},
$$

$$
\theta'_1 = \left[\frac{\nu_1}{\sigma_1^2} + (q-1)\frac{\hat{\nu}_1}{\hat{\sigma}_1^2}, ..., \frac{\nu_d}{\sigma_d^2} + (q-1)\frac{\hat{\nu}_d}{\hat{\sigma}_d^2}\right]^T,
$$

$$
\theta'_2 = -\frac{1}{2}\text{diag}\left\{\frac{1}{\sigma_1^2} + (q-1)\frac{1}{\hat{\sigma}_1^2}, ..., \frac{1}{\sigma_d^2} + (q-1)\frac{1}{\hat{\sigma}_d^2}\right\},
$$

and

$$F(\theta) = \frac{1}{2}\mu^T \Sigma^{-1}\mu + \frac{1}{2}\log(2\pi)^d|\Sigma| = \sum_{i=1}^{d}\left\{\frac{\nu_i^2}{2\sigma_i^2} + \frac{\log 2\pi}{2} + \log\sigma_i\right\},$$

$$F(\hat{\theta}) = \frac{1}{2}\hat{\mu}^T \hat{\Sigma}^{-1}\hat{\mu} + \frac{1}{2}\log(2\pi)^d|\hat{\Sigma}| = \sum_{i=1}^{d}\left\{\frac{\hat{\nu}_i^2}{2\hat{\sigma}_i^2} + \frac{\log 2\pi}{2} + \log\hat{\sigma}_i\right\},$$

$$F(\theta + (q-1)\hat{\theta}) = -\frac{1}{4}(\theta_1')^T(\theta_2')^{-1}\theta_1' + \frac{1}{2}\log|-\pi(\theta_2')^{-1}|$$

$$= \sum_{i=1}^{d}\left\{\frac{1}{2}\frac{\left(\frac{\nu_i}{\sigma_i^2} + (q-1)\frac{\hat{\nu}_i}{\hat{\sigma}_i^2}\right)^2}{\frac{1}{\sigma_i^2} + (q-1)\frac{1}{\hat{\sigma}_i^2}} + \frac{\log 2\pi}{2} + \log\frac{1}{\frac{1}{\sigma_i^2} + (q-1)\frac{1}{\hat{\sigma}_i^2}}\right\}.$$

## J  EXPERIMENT SETTING

### J.1  POLICY REGULARIZERS IN EXPERIMENTS

Table 1: Policy regularizers $\phi$ and their corresponding $f_\phi$ (Yang et al., 2019).

| reg. type. | condition | $\phi(x)$ | $f'_\phi(x)$ |
|---|---|---|---|
| Shannon | - | $-\log x$ | $-\log x - 1$ |
| Tsallis | $k > 0, q > 1$ | $\frac{k}{q-1}(1 - x^{q-1})$ | $\frac{k}{q-1}(1 - qx^{q-1})$ |
| Exp | $k \geq 0, q \geq 1$ | $q - x^k q^x$ | $q - x^k q^x(k + 1 + x \log q)$ |
| Cos | $0 < \theta \leq \pi/2$ | $\cos(\theta x) - \cos(\theta)$ | $-\cos(\theta) + \cos(\theta x) - \theta x \sin(\theta x)$ |
| Sin | $0 < \theta \leq \pi/2$ | $\sin(\theta) - \sin(\theta x)$ | $\sin(\theta) - \sin(\theta x) - \theta x \cos(\theta x)$ |

### J.2  DENSITY-BASED MODEL

By exploiting the knowledge on the reward in **Corollary 1**

$$-f'_\phi(\pi(a|s)) - \mathbb{E}_{a' \sim \pi(\cdot|s)}[f'_\phi(\pi(a'|s)) - \phi(\pi(a'|s))],$$

we consider the density-based model (DBM) which is defined by

$$r_\theta(s, a) \approx -f'_\phi(\pi_{\theta_1}(a|s)) + B_{\theta_2}(s)$$

for $\theta = (\theta_1, \theta_2)$. Here, $f'_\phi$ is a function that can be known priorly, and $\pi_{\theta_1}$ is a neural network which is defined separately from the policy neural network $\pi_\psi$. The DBM for discrete control problems are depicted in Figure 6 (*Left*). The model outputs rewards over all actions in parallel, where softmax is used for $\pi_{\theta_1}(\cdot|s)$ and $-f'_\phi$ is elementwisely applied to those softmax outputs followed by elementwisely adding $B_{\theta_2}(s)$. For continuous control (Figure 6, *Right*), we use the network architecture similar to that in discrete control, where a multivariate Gaussian distribution is used instead of a softmax layer.

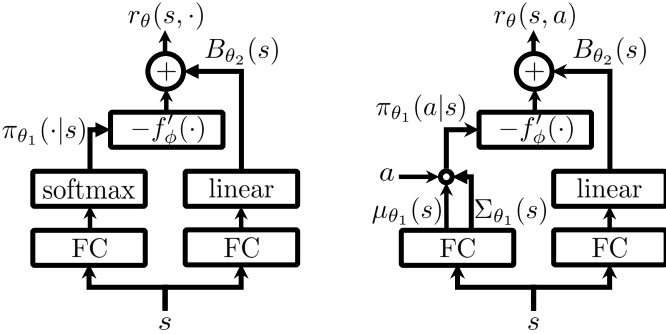

Figure 6: Density-based model for discrete (*Left*) and continuous control (*Right*)

### J.3  EXPERT IN BERMUDA WORLD ENVIRONMENT

We assume a stochastic expert defined by

$$\pi_E(a|s) = \frac{\sum_{t=1}^3 (d^{(t)})^{-1} \mathbb{I}\{a = \mathrm{Proj}(\theta^{(t)})\}}{\sum_{t=1}^3 (d^{(t)})^{-1}},$$

$$\theta^{(t)} = \arctan2(\bar{y}^{(t)} - y, \bar{x}^{(t)} - x), d^{(t)} = \|\bar{s}^{(t)} - s\|_2^4 + \epsilon, t = 1, 2, 3,$$

for $s = (x, y)$, $\bar{s}^{(1)} = (\bar{x}^{(1)}, \bar{y}^{(1)}) = (-5, 10)$, $\bar{s}^{(2)} = (\bar{x}^{(2)}, \bar{y}^{(2)}) = (0, 10)$, $\bar{s}^{(3)} = (\bar{x}^{(3)}, \bar{y}^{(3)}) = (5, 10)$, $\epsilon = 10^{-4}$ and an operator $\text{Proj}(\theta) : \mathbb{R} \to \mathcal{A}$ that maps $\theta$ to the closest angle in $\mathcal{A}$. In Figure 7, we depicted the expert policy.

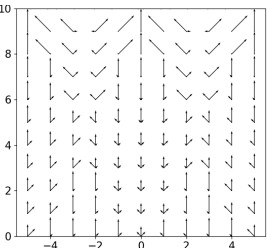

Figure 7: Visualization of the expert policy

## J.4 MUJOCO EXPERIMENT SETTING

Instead of directly using MuJoCo environments with $\tanh$-squashed policies proposed in Soft Actor-Critic (SAC) (Haarnoja et al., 2018), we move $\tanh$ to a part of the environment—named *hyperbolized environments* in short—and assume Gaussian policies. Specifically, after an action $a$ is sampled from the policies, we pass $\tanh(a)$ to the environment. We then consider multi-variate Gaussian policy

$$\pi(\cdot|s) = \mathcal{N}\left(\boldsymbol{\mu}(s), \boldsymbol{\Sigma}(s)\right)$$

with $\boldsymbol{\mu}(s) = [\mu_1(s), ..., \mu_d(s)]^T$, $\boldsymbol{\Sigma}(s) = \text{diag}\{(\sigma_1(s))^2, ..., (\sigma_d(s))^2\}$, where

$$-\text{arctanh}(0.99) \leq \mu_i(s) \leq \text{arctanh}(0.99), \log(0.01) \leq \log \sigma_i(s) \leq \log(2)$$

for all $i = 1, ..., d$. Instead of using clipping, we use $\tanh$-activated outputs and scale them to fit in the above ranges, which empirically improves the performance. Also, instead of using potential-based reward shaping used in AIRL (Fu et al., 2018), we update the moving mean of intermediate reward values and update the value network with mean-subtracted rewards—so that the value network gets approximately mean-zero reward—to stabilize the RL part of RAIRL. Note that this is motivated by **Lemma 2** from which we can guarantee that any constant shift of reward functions does not change optimality.

## J.5 HYPERPARAMETERS

Table 2, Table 3 and Table 4 list the parameters used in our Bandit, Bermuda World, and MuJoCo experiments, respectively.

Table 2: Hyperparameters for Bandit environments.

| Hyper-parameter | Bandit |
|---|---|
| Batch size | 500 |
| Initial exploration steps | 10,000 |
| Replay size | 500,000 |
| Target update rate ($\tau$) | 0.0005 |
| Learning rate | 0.0005 |
| $\lambda$ | 5 |
| $q$ (Tsallis entropy $\mathcal{T}_q^k$) | 2.0 |
| $k$ (Tsallis entropy $\mathcal{T}_q^k$) | 1.0 |
| Number of trajectories | 1,000 |
| Reward learning rate | 0.0005 |
| Steps per update | 50 |
| Total environment steps | 500,000 |

Table 3: Hyperparameters for Bermuda World environment.

| Hyper-parameter | Bermuda World |
|---|---|
| Batch size | 500 |
| Initial exploration steps | 10,000 |
| Replay size | 500,000 |
| Target update rate ($\tau$) | 0.0005 |
| Learning rate | 0.0005 |
| $q$ (Tsallis entropy $\mathcal{T}_q^k$) | 2.0 |
| $k$ (Tsallis entropy $\mathcal{T}_q^k$) | 1.0 |
| Number of trajectories | 1,000 |
| Reward learning rate | 0.0005 |
| (For evaluation) $\lambda$ | 1 |
| (For evaluation) Learning rate | 0.001 |
| (For evaluation) Target update rate ($\tau$) | 0.0005 |
| Steps per update | 50 |
| Number of steps | 500,000 |

Table 4: Hyperparameters for MuJoCo environments.

| Hyper-parameter | Hopper | Walker2d | HalfCheetah | Ant |
|---|---|---|---|---|
| Batch size | 256 | 256 | 256 | 256 |
| Initial exploration steps | 10,000 | 10,000 | 10,000 | 10,000 |
| Replay size | 1,000,000 | 1,000,000 | 1,000,000 | 1,000,000 |
| Target update rate ($\tau$) | 0.005 | 0.005 | 0.005 | 0.005 |
| Learning rate | 0.001 | 0.001 | 0.001 | 0.001 |
| $\lambda$ | 0.0001 | 0.000001 | 0.0001 | 0.000001 |
| $k$ (Tsallis entropy $\mathcal{T}_q^k$) | 1.0 | 1.0 | 1.0 | 1.0 |
| Number of trajectories | 100 | 100 | 100 | 100 |
| Reward learning rate | 0.001 | 0.001 | 0.001 | 0.001 |
| Steps per update | 1 | 1 | 1 | 1 |
| Number of steps | 1,000,000 | 1,000,000 | 1,000,000 | 2,000,000 |

