# OpenReview forum: "Regularized Inverse Reinforcement Learning"
_ICLR.cc/2021/Conference — ICLR 2021 Spotlight_

### Official Review · AnonReviewer4 · 2020-10-28
**Nice theoretical contribution but the paper would be stronger if it did more to motivate why we would want regularizers other than Shannon entropy.**

**Rating:** 6
**Confidence:** 4

**Review:**

This paper proposes a new method for regularized inverse RL. The paper builds upon work by Geist et al. who studied regularized MDPs with convex policy regularizers. The Shannon entropy is a special case of such a policy regularizer. The paper extends the analysis of Geist et al. for regularized IRL and devises tractable solutions to regularized IRL that only depend on the analytic knowledge of the regularizer. The paper further proposes regularized adversarial IRL (RAIRL), an extension of AIRL by Fu et al., as an algorithm for IRL in regularized MDPs. The algorithm is validated on a number of domains.

The paper is generally clearly written. I believe that the paper is technically correct (and I appreciated that the derivations are well explained in the appendix). The paper is novel to the best of my knowledge and I think better forms of regularization could improve the state of the art in inverse RL and will be of interest to the ICLR community.

I think it's cool that this paper allows us to solve inverse RL problems in Tsallis entropy regularized MDPs and a great technical contribution. However I think what is less clear to me is why we would want to. As it is I'm leaning towards acceptance because I enjoyed the paper and appreciate the technical contribution but I think the paper would be substantially stronger if it made a better case for why regularizers other than the normal Shannon entropy are relevant for practical IRL problems (perhaps even with an example application demonstrating a clear benefit over AIRL as well as other imitation learning baselines).

Minor comments/questions/typos:
- Would RAIRL-NSM eventually reach expert performance on the ant-v2 task?
- I find the results in figure 4 a little surprising. I would have expected the divergence to be minimal for q=q'. Could you comment further on this empirical result?
- I'm wondering why the exponential/cosine/sine regularizers were chosen in experiment 1. Are these just meant to demonstrate the flexibility of the framework or is there a motivation for using them in practice?
- I appreciate the need to stay within the page limit but would encourage the authors to decompress the paper a bit for a potential camera-ready version.
- I think it would be nice to include an algorithm box as a summary of the proposed algorithm.
- Figures 3 and 4 are a bit small and hard to parse.
- The paper might benefit from some further proofreading for typos and small grammatical errors such as missing articles

---

> ### Author Response · Authors · 2020-11-20
> **Response to Reviewer 4**
>
> **I think what is less clear to me is why we would want to. As it is I'm leaning towards acceptance because I enjoyed the paper and appreciate the technical contribution but I think the paper would be substantially stronger if it made a better case for why regularizers other than the normal Shannon entropy are relevant for practical IRL problems (perhaps even with an example application demonstrating a clear benefit over AIRL as well as other imitation learning baselines).**
>
> Although we agree with your concerns, we made our contributions more focused on theoretical and algorithmic generalizations in regularized IRL at this submission, since those generalizations already require a tremendous amount of mathematical derivation and experiments. However, we remain choosing appropriate regularizers for practical benefits as our future direction.
>
> Bregman divergence has widely appeared in the history of machine learning, and there’s also a recent work that found the practical benefit of considering Bregman divergence in a classification problem. (Ehsan et al., "Robust Bi-Tempered Logistic Loss Based on Bregman Divergences". NeurIPS 2019) Briefly, motivated from Bregman divergence, this work modifies the loss function to manage the sensitivity of the classification boundary w.r.t. training samples. We believe the similar approach can be done to improve the robustness of IRL w.r.t. noisy expert demonstrations or outliers, but we left this as our future work.
>
> **Would RAIRL-NSM eventually reach expert performance on the ant-v2 task?**
>
> This is an open question, but we’ll try to figure this out within the discussion period.
>
> **I find the results in figure 4 a little surprising. I would have expected the divergence to be minimal for q=q'. Could you comment further on this empirical result?**
>
> In our example in Figure 1, you can see that Tsallis entropy regularizer with higher $q$ requires much tighter distribution matching via Bregman divergence minimization. Importantly and Interestingly, we could observe the size of the valley becomes smaller around the green point (the expert policy). We use Gaussian policies in our MuJoCo experiments and believe analogous results are observed. That is, minimizing Bregman divergence with higher $q$ minimizes the divergence with lower $q$ as well.
>
> **I'm wondering why the exponential/cosine/sine regularizers were chosen in experiment 1. Are these just meant to demonstrate the flexibility of the framework or is there a motivation for using them in practice?**
>
> This was basically to show the flexibility and generalizability of RAIRL and to review our mathematical derivations and experiments by referring Yang et al. 2019.
>
> **I appreciate the need to stay within the page limit but would encourage the authors to decompress the paper a bit for a potential camera-ready version.**
>
> With the additional one page allowed during the rebuttal period, we reorganize figures and the algorithm box and add some additional information on experimental settings for clarification.
>
> **I think it would be nice to include an algorithm box as a summary of the proposed algorithm.**
>
> The algorithm box was originally given in the appendix due to the page limitation. We move it to the main part of our revision.
>
> **Figures 3 and 4 are a bit small and hard to parse.**
>
> We increase the size of figures to improve the readability in our revision.
>
> **The paper might benefit from some further proofreading for typos and small grammatical errors such as missing articles.**
>
> We added some references from other reviewers and clarify the meaning of tractability in our revision. We also added **our new finding in the appendix**---why original solution in Geist et al is intractable/how the original solution is related to ours.

---

> > ### Comment · AnonReviewer4 · 2020-11-23
> > **Thanks for the clarifications.**
> >
> > I would like to thank the authors for their answers to my questions. I will update my review after the end of the discussion period.

---

### Official Review · AnonReviewer1 · 2020-10-29

**Rating:** 7
**Confidence:** 3

**Review:**

=====POST-REBUTTAL COMMENTS========

I thank the authors for the response and the efforts in the updated draft. I think the paper is stronger and should be accepted.


##########################################################################
Summary:
This paper examines the problem of regularized IRL. As opposed to standard IRL, which can have degenerate solutions, regularized IRL has a unique optimal solution. The authors examine different forms of regularization and derive an efficient IRL algorithm which generalizes AIRL and is applicable to continuous control tasks.

##########################################################################
Reasons for score:
The  idea of regularized IRL is a nice contribution to the field. It ties in nicely with recent work on regularized RL and overcomes some of the challenges of classical IRL. The mathematical foundation is rigorous and the experimental results are promising.

##########################################################################
Pros: 1. Nice general mathematical framework that generalizes prior work. 2. Tractable algorithms that work in continuous state and action domains. 3. The empirical analysis of the actual divergence is interesting.

##########################################################################
Cons: 1. The paper is a bit notation heavy and that makes it hard to follow. 2. Some undefined notation such as \mathcal{D} on page 2.  3. Lacking in justification for the choice of entropy regularizers. Why Tsallis? Why follow Yang 2019 in choice of regularizer? 4. In general a lot of the theoretical results are given without any intuitive explanation for what they mean or how to interpret them. 5. It's unclear how this is different from the theory for GAIL which is also in terms of a general regularizer.

##########################################################################
Other comments:

The sentences before the discussion mention that RAIRL is good for safe imitation learning. This is unclear since safety is undefined. Also in the same sentences the authors mention that RAIRL fails to minimize divergence, but it seems to work in the figure for some values of q'. What do the authors mean by fail?

Figure 1 seems to transition from having high divergence in the top to low divergence (for  small log \sigma). Why?

Rather than using a single dash for parentheticals, it is usually better to use latex em-dash for parentheticals by using three dashes --- with no space before and after

---

> ### Author Response · Authors · 2020-11-20
> **Response to Reviewer 1**
>
> **The paper is a bit notation heavy and that makes it hard to follow.**
>
> At this revision, we think $\pi$ was sometimes ambiguously used and confusing, and thus, clarify the notions by using $p$ to indicate arbitrary distributions on $\mathcal{A}$. We’ll keep updating our submission for mathematical clarification.
>
> **Some undefined notation such as $\mathcal{D}$ on page 2.**
>
> This was a typo. We fixed it in our revision.
>
> **Lacking in justification for the choice of entropy regularizers. Why Tsallis? Why follow Yang 2019 in the choice of regularizer?**
>
> In Yang et al. 2019, RL with Tsallis/exp/cos/sin regularizers has shown to perform comparably with Shannon-entropy-regularized RL. Other regularizers haven’t been empirically validated to the best of our knowledge. We chose those regularizers to make our empirical analysis more tractable and reliable. A broader comparison among regularizers remains as our future work. One approach is to define and think of robustness in IRL, as [Amid et al. 2019](https://papers.nips.cc/paper/2019/file/8cd7775f9129da8b5bf787a063d8426e-Paper.pdf) did for supervised learning problems by using the Bregman divergence.
>
> **In general, a lot of the theoretical results are given without any intuitive explanation for what they mean or how to interpret them.**
>
> **Lemma 1** shows one of the tractable solutions in regularized IRL, whereas the previous work (Geist et al.) required a closed-form relation between optimal policy and value function. **Lemma 2** says other solutions can be found by shaping the reward in **Lemma 1**, which we believe is already stated in our submission. Finally, based on R1’s review, we think **Lemma 3** and its motivation may have confused R1 a bit. For a more concrete explanation on this, please note that the *state-action visitation distributions* and their matching have contributed to the recent advance in imitation learning, but our **Lemma 1**/**Corollary1**/**Lemma 2** are not derived from that perspective. In **Lemma 3** of our submission, we show that RL with our proposed rewards can be regarded as the problem of matching state-action visitation distributions under some conditions. We update our revision to clarify our intuitions.
>
> **It's unclear how this is different from the theory for GAIL which is also in terms of a general regularizer.**
>
> GAIL’s theorems are not directly related to our **Lemma 3**, but some terms are related. To be specific, when the regularizer $\Omega(p)=-H(p)$ (Shannon entropy regularizer) is used for $p\in\Delta^{\mathcal{A}}$, the term  $\bar{\Omega}$ in **Lemma 3** of our submission is equal to $-\bar{H}$ in **Lemma 3.1** of GAIL. Since $-\bar{H}$ is strictly convex from **Lemma 3.1** of GAIL, we can say Shannon-entropy-regularized RL with a reward in $\log\pi_E(a|s)$ (from **Lemma 1** of our submission) is equivalent to the minimization of Bregman divergence defined by $-\bar{H}$. With **Theorem 3** of Lee et al. 2018, a similar approach can be done for Tsallis entropy regularizer. We clarify this in our revision.
>
> **The sentences before the discussion mention that RAIRL is good for safe imitation learning. This is unclear since safety is undefined. Also in the same sentences, the authors mention that RAIRL fails to minimize divergence, but it seems to work in the figure for some values of q'. What do the authors mean by fail?**
>
> We wanted to emphasize that higher $q$ leads to tighter distribution matching, but we agree that safety is undefined and ambiguously used.  We remove the sentence regarding safety.
>
> Also, we clarify RAIRL’s training and (reward) evaluation in our revision.
> During RAIRL’s training, we found mean Bregman divergences are minimized as we desired, whereas during the reward evaluation---RAC with a randomly initialized agent and the reward acquired from RAIRL---doesn’t seem to properly minimize mean Bregman divergence. We speculate this result mainly comes from the fact that probability density is used for the agent’s policy $\pi$ in continuous control problems; each density is unbounded in principle and may have a much larger value than its value in discrete control problems where probability mass function is used. Solving this problem and developing more practical algorithms (relative to SOTA imitation learning methods) are remained as future work.
>
> **Figure 1 seems to transition from having high divergence in the top to low divergence (for small $\log\sigma$). Why?**
>
> The divergence is plotted for $\pi_E$ fixed at the green point of Figure 1 while we vary the mean and standard deviation of $\pi$. Therefore, the divergence has a lower value around the green point.
>
> **Rather than using a single dash for parentheticals, it is usually better to use latex em-dash for parentheticals by using three dashes --- with no space before and after.**
>
> We appreciate your comment. We use three dashes in our revision.

---

### Official Review · AnonReviewer2 · 2020-10-29
**Regularized Inverse Reinforcement Learning**

**Rating:** 6
**Confidence:** 3

**Review:**

Pros:
1. this paper studies an interesting problem called regularized inverse reinforcement learning, which is novel to me and brings me some knowledge.
2. this paper first proposes a general solution for regularized IRL; then, TSALLIS entropy is proposed with IRL, as well as an adversary learning-based training strategy. The methodology part seems reasonable and the contribution is good.
3. the paper is well-organized. The experiments are convincing, trying to illustrate the performance under different scenarios, including continuous and discrete reinforcement learning settings.

Cons:
1. in the experiments, much comparison is internal comparison of proposed methods. In the beginning, the authors mention Shannon-entropy regularizers' limitations. The authors should conduct more experiments to prove the statement. Now, only Experiment 3 mentions it.
2. I don't see many comparisons with other baselines. Adding more baselines is better.


In summary, this paper studies a novel problem, regularized inverse reinforcement learning. The paper proposes several techniques to solve the regularization in different aspects. The experiments are conducted under different settings, but some cons are needed to revise.

---

> ### Author Response · Authors · 2020-11-20
> **Response to Reviewer 2**
>
> **In the experiments, much comparison is an internal comparison of proposed methods. In the beginning, the authors mention Shannon-entropy regularizers' limitations. The authors should conduct more experiments to prove the statement. Now, only Experiment 3 mentions it.**
>
> We’d like to emphasize that both our motivation and contribution are on the generalized perspective on IRL, not on outperforming the baselines with Shannon-entropy regularization. (Please note that we have not mentioned any restriction regarding Shannon entropy regularizer.) In experiment 3, we wanted to show that RAIRL performs on par regardless of $q$, which shows RAIRL’s capability of generalization.
>
> **I don't see many comparisons with other baselines. Adding more baselines is better.**
>
> In principle, RAIRL with Shannon entropy regularizer is equivalent to AIRL (Fu et al., 2018). Please note that we included Shannon-entropy regularizer for all of our experiments. Other imitation learning baselines---not IRL baselines---such as Discriminator Actor-Critic (DAC, Kostrikov et al., 2019) or GAIL (Ho et al., 2016) are not considered in our submission since
> 1. they consider imitation learning and bypass reward learning
> 2. we did not focus on outperforming the baselines and seeking SOTA performance, but on the generalized perspective on IRL and AIRL (algorithmic generalization).
>
> Additionally, although we haven’t launched the experiments with DAC and GAIL, we believe the scores we reported in our submission are sufficient to compare with those in the [DAC paper](https://arxiv.org/abs/1809.02925) (See Figure 8 in Appendix B of DAC for MuJoCo benchmarks). In summary, we guess RAIRL will outperform GAIL and slightly underform DAC.

---

### Official Review · AnonReviewer5 · 2020-11-06
**Good paper that generalizes policy regularization in regularized MDPs**

**Rating:** 8
**Confidence:** 4

**Review:**

This paper shows a formulation of regularized Markov Decision Processes (MDPs), which is slightly different from that of Geist et al. (2019). Then, the authors propose a novel inverse reinforcement learning under regularized MDPs. One of the contributions is that policy regularization considered here is more general than that of Yang et al. (2019).

This paper is written very well and is of publishing quality. I think it is sufficiently significant to be accepted. Still, I have the following questions.

1. The proposed method is based on the relationship between imitation learning and statistical divergence minimization. If my understanding is correct, Bregman divergence plays a role in generalizing generalized adversarial imitation learning. However, as the authors mentioned in Section 6, Bregman divergence does not include f-divergence, which is also studied in imitation learning. Would you discuss the connection to the formulation using f-divergence in more detail?

2. I am interested in the relationship between the proposed method and Lee et al. (NeurIPS2018). Is the proposed method nearly the same as Lee et al. (2018) when Tsallis entropy is selected as regularization? If not, does the proposed method outperform Lee et al. (2018) in the MuJoCo control tasks?

3. The authors claim that the solutions provided by Geist et al. (2019) are intractable in the Introduction. However, it is shown that the reward baseline term in Corollary 1 is intractable except for some well-studied setups. Does it imply that the proposed method faces the same difficulty when applied with arbitrary policy regularization?

4. The experimental results shown in Figure 3 is interesting, but I have a few concerns. In some cases, the averaged Bregman divergence of RAIRL-NSM (\lambda = 1) was larger than that of Random. Would you show the example of the learned policy for the readers’ understanding? Besides, is the same policy regularization used in Behavior Cloning? Finally, are exp, cos, and sin the meaningful regularizer?

5. To derive the practical algorithms, the authors consider the same form of the policy regularization used by Yang et al. (2019), which is given by - \lambda E[\phi(\pi(a))]. Is it possible to derive the algorithm in which the regularizer is given by \Omega(\pi)?

---

> ### Author Response · Authors · 2020-11-20
> **Response to Reviewer 5 (2/2)**
>
> **The experimental results shown in Figure 3 is interesting, but I have a few concerns. In some cases, the averaged Bregman divergence of RAIRL-NSM ($\lambda = 1$) was larger than that of Random. Would you show the example of the learned policy for the readers’ understanding?**
>
> What we measure is the Bregman divergence between the agent and the expert policies, it can be larger than the Bregman divergence between uniform and the expert policies. For example, let us consider KL divergence $D_{KL}(\pi||\pi_E)=\sum_{a}\pi(a)\log\frac{\pi(a)}{\pi_{_E}(a)}$ for $a=0$ or $1$ with $\pi_E(0)=0.1, \pi_E(1)=0.9$. When $\pi$ is a uniform distribution, i.e., $\pi(0)=\pi(1)=0.5$, KL divergence is equal to $0.5\times\log(0.5/0.1)+0.5\times\log(0.5/0.9)\approx0.510826$. But if $\pi(0)=1.0, \pi(1)=0.0$---that is, the probability mass is more on non-expert-like behavior. KL divergence is equal to $1.0\times\log(1.0/0.1)\approx2.30259$ which is higher than that of uniform policy. We believe such a result happens when there’s an error for reward learning. For example, RAIRL-NSM failed to recover the ground truth reward in the multi-armed bandit experiment in Figure 2, especially for sparse expert policy. The expert in the Bermuda environment is sparse as well, and we can think the similar situation happened for RAIRL-NSM.
>
> **Besides, is the same policy regularization used in Behavior Cloning?**
>
> In Behavioral Cloning, we use supervised learning loss (cross entropy loss) and policy regularizer is not considered.
>
> **Finally, are exp, cos, and sin the meaningful regularizer?**
>
> Yang et al. (2019) devised those regularizer to achieve sparse optimal policies---we get dense optimal policy for Shannon entropy regularizer---and has empirically shown that it works.
>
> **To derive the practical algorithms, the authors consider the same form of the policy regularization used by Yang et al. (2019), which is given by -$\lambda E[\phi(\pi(a))]$. Is it possible to derive the algorithm in which the regularizer is given by $\Omega(\pi)$?**
>
> Our algorithm can be used with a general $\Omega$, as long as we can evaluate $\Omega, \nabla\Omega$ and the expectation over $\nabla\Omega$, although our experiments were restricted to the regularizers from Yang et al. (2019).(2019), which is given by -$\lambda E[\phi(\pi(a))]$.

---

> ### Author Response · Authors · 2020-11-20
> **Response to Reviewer 5 (1/2)**
>
> **If my understanding is correct, Bregman divergence plays a role in generalizing generalized adversarial imitation learning. However, as the authors mentioned in Section 6, Bregman divergence does not include $f$-divergence, which is also studied in imitation learning. Would you discuss the connection to the formulation using $f$-divergence in more detail?**
>
> To be correct, we use Bregman divergence to generalize MaxEntIRL, not GAIL, and propose RAIRL to extend AIRL which was motivated by MaxEntIRL.
>
> Regarding our comments in Section 6, here’s specifically what we want to say in our submission. What we mainly used in our submission is that the regularized RL with our proposed solution is equivalent to minimizing **the discounted sum over Bregman divergences between policies**. Since the $f$-divergence is generally not in a class of Bregman divergence, it’s not possible to get **the discounted sum over $f$-divergences between policies**, which will also give us the expert policy at the optimality, if we confine ourselves in a regularized MDP framework of Geist et al. In other words, we need to devise an MDP framework which is different from a regularized MDP of Geist et al. to end up with the discounted sum over $f$-divergences between policies.
>
> Regarding the $f$-divergences in existing works---where those are motivated by f-GAN paper---please note that those divergences are defined not for the policies, but for state-action visitation distributions (occupancy measures). Although what we wanted to mention in Section 6 wasn’t actually related to those previous works, we believe **Lemma 3** of our submission can answer your question in terms of state-action visitation distributions. Briefly, RL with our solution can be seen as minimizing **the Bregman divergence between visitation distributions** when some condition for $\bar{\Omega}$ is satisfied. Therefore, we can say the existing works on $f$-divergence minimization in imitation learning have exploited totally different ways.
>
> **I am interested in the relationship between the proposed method and Lee et al. (NeurIPS2018). Is the proposed method nearly the same as Lee et al. (2018) when Tsallis entropy is selected as regularization? If not, does the proposed method outperform Lee et al. (2018) in the MuJoCo control tasks?**
>
> Both maximum causal Tsallis entropy imitation learning (MCTEIL, Lee et al., 2018) and RAIRL consider Tsallis entropy regularizer, but there are differences in motivations.
>
> MCTEIL was focused on $q=2$, which enables us to avail the closed-form derivation of Tsallis entropy for the mixture of Gaussian policies and the corresponding richer expressive power. On the other hand, we consider a single Gaussian policy and the theorem applicable to arbitrary $q$.
>
> Although we haven’t explicitly compared the performance---since (1) outperforming the baselines is not the objective of our submission and (2) it’s not available to directly compare the results due to the use of a mixture of Gaussian policies in MCTEIL---we speculate MCTEIL with single Gaussian policy performs on par with GAIL (with TRPO, on-policy RL), and thus, will underperform RAIRL (with RAC, off-policy RL) in imitation learning.
>
> We believe the scores we reported in our submission are sufficient to compare with those in the [DAC paper](https://arxiv.org/pdf/1809.02925.pdf) (See Figure 8 in Appendix B of DAC for MuJoCo benchmarks). We guess RAIRL will outperform GAIL and slightly underperform DAC.
>
> **The authors claim that the solutions provided by Geist et al. (2019) are intractable in the Introduction. However, it is shown that the reward baseline term in Corollary 1 is intractable except for some well-studied setups. Does it imply that the proposed method faces the same difficulty when applied with arbitrary policy regularization?**
>
> We specify this part in our revision through Appendix F.1 and F.2. Briefly, the original solution in Appendix F.1 requires far more additional knowledge (on model dynamics) and derivation (for the convex conjugate $\Omega^*$ of $\Omega$), which prevents us to use that reward in practice. On the other hand, when $\Omega$ and $\nabla\Omega$ can be evaluated---which can be simply derived from $\Omega$---we can always calculate our solution in Lemma 1 for discrete control. For continuous control, the tractability for the expectation (reward baseline) is additionally required to calculate our solution, which can be done for Tsallis entropy as shown in our submission.

---

### Official Review · AnonReviewer3 · 2020-11-10
**Requesting clarifications on methodology and some other components**

**Rating:** 7
**Confidence:** 3

**Review:**

This work considers a regularized IRL setup, where instead of the entropy regularization used in maximum entropy IRL, an arbitrary convex regularizer $\Omega$ is used. The work presents a number of theoretical results for this general setting, and it is shown that when $\Omega$ is Tsallis entropy, the $RL \cdot IRL$ is equivalent to minimizing a Bregman divergence defined based on the Tsallis entropy and the expert state-action distribution. A practical algorithm is presented for IRL with the Tsallis entropy. A number of experiments are performed to obtain understanding of various components.

I hope that the following questions can be resolved during the discussion period as some things are a bit unclear to me which is preventing me from providing a more detailed analysis of the work.

* Section 3
  * Section 3.1, paragraph after equation 6: Can you clarify (for example with concrete examples or equations) what you mean by terms like "functional-form" and "intractable"?
* Section 4
  * __Could you clarify your structured discriminator? What exactly are you using for $t(s,a,\pi)$? How does this relate to the Tsallis entropy baseline equations derived earlier?__
  * Equation 13: Should this be argmax? I think maybe this should instead be just a small gradient update using this objective since $\hat{t}(s,a;\pi^{(i)})$ is the correct objective locally around $\pi^{(i)}$?
* Section 5
  * __Based on the description in H.2 I don't understand what the Density-Based Model is. Please clarify.__
  * Section 5.1: Can you clarify what you mean by "acquires the ground truth rewards"? Figure 2 is showing different reward curves for each method, so how can they all be the ground truth?
  * Section 5.2: Can you explain how you are computing the divergences between the joint state-action distributions $p(s,a)$ of the expert and trained policy? Also how do you compute these divergences for the Mujoco experiments in section 5.3?
  * Section 5.3: Could you elaborate what you mean by "Unfortunately, RAIRL fails to acquire a reward function that effectively minimizes the target divergence in continuous controls."?

You may also want to cite:
@article{ke2019imitation,
  title={Imitation Learning as $ f $-Divergence Minimization},
  author={Ke, Liyiming and Barnes, Matt and Sun, Wen and Lee, Gilwoo and Choudhury, Sanjiban and Srinivasa, Siddhartha},
  journal={arXiv preprint arXiv:1905.12888},
  year={2019}
}

---

> ### Author Response · Authors · 2020-11-20
> **Response to Reviewer 3**
>
> **Section 3.1, paragraph after equation 6: Can you clarify (for example with concrete examples or equations) what you mean by terms like "functional-form" and "intractable"?**
>
> Since the terms like “analytical”, “functional-form”, “tractable”, “intractable” in our submission are a bit ambiguously defined and used, we update our revision to only use either tractable or intractable and clearly define the tractable reward function in Appendix F.1. Also, relevant contents in the main part are updated correspondingly. We also add how the original solution in Geist et al. is related to our solution in Appendix F.2, **which is a new finding during this rebuttal period**.
>
> **Section 4: Could you clarify your structured discriminator? What exactly are you using for $t(s, a; \pi)$? How does this relate to the Tsallis entropy baseline equations derived earlier?**
>
> For $t(s, a; \pi)$, we use the formula in Eq. (9) of **Corollary 1**. To be specific, for $\pi^{(i)}$ of the $i$-th iteration, we use $t(s, a; \pi^{(i)})$ for discriminator training. Please note that the functions related to the regularizer in **Corollary 1**---$f_\phi’$ and $\phi$---are priorly known and can be evaluated in regularized IRL. Therefore for the discrete control problems, we can always evaluate $t(s, a; \pi^{(i)})$, whereas in continuous control, the expectation (the reward baseline) is intractable depending on the density of the policy---except for Shannon entropy regularizer where the reward baseline=-1 regardless of the density. However, for Tsallis entropy regularizer with multivariate Gaussian policy, we show that a tractable form of reward baseline can be found, and we use this for our experiments with continuous controls.
>
> **Equation 13: Should this be argmax? I think maybe this should instead be just a small gradient update using this objective since $t(s, a;\pi^{(i)})$ is the correct objective locally around $\pi^{(i)}$?**
>
> It doesn't have to be argmax and the intuitive idea we had is exactly the same as what you mentioned. We updated the relevant part accordingly.
>
> **Section 5: Based on the description in H.2 I don't understand what the Density-Based Model is. Please clarify.**
>
> We clarify density-based model with additional figure in Appendix J.2 in our revision.
>
> **Section 5.1: Can you clarify what you mean by "acquires the ground truth rewards"? Figure 2 is showing different reward curves for each method, so how can they all be the ground truth?**
>
> In the experiment with multi-armed bandit, we know the experts’ policy $\pi_E$ priorly. Therefore, we can explicitly evaluate $t(s, a;\pi_E)$ (called ground truth reward since it is reward learning objective of RAIRL) by using $\pi_E$ and the equation in **Corollary 1**.
>
> **Section 5.2: Can you explain how you are computing the divergences between the joint state-action distributions $p(s, a)$ of the expert and trained policy? Also, how do you compute these divergences for the Mujoco experiments in section 5.3?**
>
> For both discrete and continuous controls, the Bregman divergence can be written as $D_\Omega^{\mathcal{A}}(p_1||p_2)=E_{a\sim p_1}[f_\phi'(p_2(a))-\phi(p_1(a))]-E_{a\sim p_2}[f_\phi'(p_2(a))-\phi(p_2(a))]$ for distributions $p_1=\pi(\cdot|s)$ (the agent policy) and $p_2=\pi_E(\cdot|s)$ (the expert policy), and we can evaluate the Bregman divergence on each state since we know both policies priorly. We report mean Bregman divergence over states, where those states are coming from 30 evaluation trajectories.
>
> **Section 5.3: Could you elaborate on what you mean by "Unfortunately, RAIRL fails to acquire a reward function that effectively minimizes the target divergence in continuous controls."?**
>
> We clarify RAIRL’s training and (reward) evaluation in our revision. During RAIRL’s training, we found the mean Bregman divergences are minimized as we desired, whereas during the reward evaluation---RAC with a randomly initialized agent and the reward acquired from RAIRL---RAIRL doesn’t seem to properly minimize mean Bregman divergence. We speculate this result mainly comes from the fact that probability density is used for the agent’s policy $\pi$ in continuous control problems; each density is unbounded in principle and may have much larger value than its value in discrete control problems where probability mass function is used. We left the solution of this open problem (relative to SOTA imitation learning methods) as our future works.
>
> **You may also want to cite: @article{ke2019imitation, title={Imitation Learning as $f$-Divergence Minimization}, author={Ke, Liyiming and Barnes, Matt and Sun, Wen and Lee, Gilwoo and Choudhury, Sanjiban and Srinivasa, Siddhartha}, journal={arXiv preprint arXiv:1905.12888}, year={2019} }**
>
> We add this to our reference.

---

### Decision · Program_Chairs · 2021-01-07
**Final Decision**

**Decision:**

Accept (Spotlight)

**Comment:**

This paper studies inverse reinforcement learning through the prism of regularized Markov decision processes, by generalizing MaxEntIRL from the negative entropy to any strongly convex regularizer (as a side note, strict convexity might be enough for many results).
The reviewers appreciated the clarity, the mathematical rigor and the empirical evaluation of this paper. They asked some questions and raised some concerns, that were mostly addressed in the rebuttal and the revision provided by the authors.
This is a strong paper, for which the AC recommends acceptance.